# Asynchronous Parallel Stochastic Gradient for Nonconvex Optimization

**Xiangru Lian, Yijun Huang, Yuncheng Li, and Ji Liu**
Department of Computer Science, University of Rochester
{lianxiangru,huangyj0,raingomm,ji.liu.uwisc}@gmail.com

## Abstract

Asynchronous parallel implementations of stochastic gradient (SG) have been broadly used in solving deep neural network and received many successes in practice recently. However, existing theories cannot explain their convergence and speedup properties, mainly due to the nonconvexity of most deep learning formulations and the asynchronous parallel mechanism. To fill the gaps in theory and provide theoretical supports, this paper studies two asynchronous parallel implementations of SG: one is over a computer network and the other is on a shared memory system. We establish an ergodic convergence rate $O(1/\sqrt{K})$ for both algorithms and prove that the linear speedup is achievable if the number of workers is bounded by $\sqrt{K}$ ($K$ is the total number of iterations). Our results generalize and improve existing analysis for convex minimization.

## 1  Introduction

The asynchronous parallel optimization recently received many successes and broad attention in machine learning and optimization [Niu et al., 2011, Li et al., 2013, 2014b, Yun et al., 2013, Fercoq and Richtárik, 2013, Zhang and Kwok, 2014, Marecek et al., 2014, Tappenden et al., 2015, Hong, 2014]. It is mainly due to that the asynchronous parallelism largely reduces the system overhead comparing to the synchronous parallelism. The key idea of the asynchronous parallelism is to allow all workers work independently and have no need of synchronization or coordination. The asynchronous parallelism has been successfully applied to speedup many state-of-the-art optimization algorithms including stochastic gradient [Niu et al., 2011, Agarwal and Duchi, 2011, Zhang et al., 2014, Feyzmahdavian et al., 2015, Paine et al., 2013, Mania et al., 2015], stochastic coordinate descent [Avron et al., 2014, Liu et al., 2014a, Sridhar et al., 2013], dual stochastic coordinate ascent [Tran et al., 2015], and randomized Kaczmarz algorithm [Liu et al., 2014b].

In this paper, we are particularly interested in the asynchronous parallel stochastic gradient algorithm (AsySG) for *nonconvex* optimization mainly due to its recent successes and popularity in deep neural network [Bengio et al., 2003, Dean et al., 2012, Paine et al., 2013, Zhang et al., 2014, Li et al., 2014a] and matrix completion [Niu et al., 2011, Petroni and Querzoni, 2014, Yun et al., 2013]. While some research efforts have been made to study the convergence and speedup properties of AsySG for *convex* optimization, people still know very little about its properties in *nonconvex* optimization. Existing theories cannot explain its convergence and excellent speedup property in practice, mainly due to the nonconvexity of most deep learning formulations and the asynchronous parallel mechanism. People even have no idea if its convergence is certified for nonconvex optimization, although it has been used widely in solving deep neural network and implemented on different platforms such as computer network and shared memory (for example, multicore and multiGPU) system.

To fill these gaps in theory, this paper tries to make the first attempt to study AsySG for the following nonconvex optimization problem

$$\min_{x \in \mathbb{R}^n} \quad f(x) := \mathbb{E}_\xi[F(x;\xi)] \tag{1}$$

where $\xi \in \Xi$ is a random variable and $f(x)$ is a smooth (but not necessarily convex) function. The most common specification is that $\Xi$ is an index set of all training samples $\Xi = \{1, 2, \cdots, N\}$ and $F(x; \xi)$ is the loss function with respect to the training sample indexed by $\xi$.

We consider two popular asynchronous parallel implementations of SG: one is for the computer network originally proposed in [Agarwal and Duchi, 2011] and the other one is for the shared memory (including multicore/multiGPU) system originally proposed in [Niu et al., 2011]. Note that due to the architecture diversity, it leads to two different algorithms. The key difference lies on that the computer network can naturally (also efficiently) ensure the atomicity of reading and writing the *whole* vector of $x$, while the shared memory system is unable to do that efficiently and usually only ensures efficiency for atomic reading and writing on a *single* coordinate of parameter $x$. The implementation on computer cluster is described by the "consistent asynchronous parallel SG" algorithm (ASYSG-CON), because the value of parameter $x$ used for stochastic gradient evaluation is **con**sistent – an existing value of parameter $x$ at some time point. Contrarily, we use the "inconsistent asynchronous parallel SG" algorithm (ASYSG-INCON) to describe the implementation on the shared memory platform, because the value of parameter $x$ used is **incon**consistent, that is, it might not be the real state of $x$ at any time point.

This paper studies the theoretical convergence and speedup properties for both algorithms. We establish an asymptotic convergence rate of $O(1/\sqrt{KM})$ for ASYSG-CON where $K$ is the total iteration number and $M$ is the size of minibatch. The linear speedup[1] is proved to be achievable while the number of workers is bounded by $O(\sqrt{K})$. For ASYSG-INCON, we establish an asymptotic convergence and speedup properties similar to ASYSG-CON. The intuition of the linear speedup of asynchronous parallelism for SG can be explained in the following: Recall that the serial SG essentially uses the "stochastic" gradient to surrogate the accurate gradient. ASYSG brings additional deviation from the accurate gradient due to using "stale" (or delayed) information. If the additional deviation is relatively minor to the deviation caused by the "stochastic" in SG, the total iteration complexity (or convergence rate) of ASYSG would be comparable to the serial SG, which implies a nearly linear speedup. This is the key reason why ASYSG works.

The main contributions of this paper are highlighted as follows:

- Our result for ASYSG-CON generalizes and improves earlier analysis of ASYSG-CON for convex optimization in [Agarwal and Duchi, 2011]. Particularly, we improve the upper bound of the maximal number of workers to ensure the linear speedup from $O(K^{1/4}M^{-3/4})$ to $O(K^{1/2}M^{-1/2})$ by a factor $K^{1/4}M^{1/4}$;
- The proposed ASYSG-INCON algorithm provides a more accurate description than HOGWILD! [Niu et al., 2011] for the lock-free implementation of ASYSG on the shared memory system. Although our result does not strictly dominate the result for HOGWILD! due to different problem settings, our result can be applied to more scenarios (e.g., nonconvex optimization);
- Our analysis provides theoretical (convergence and speedup) guarantees for many recent successes of ASYSG in deep learning. To the best of our knowledge, this is the first work that offers such theoretical support.

**Notation** $x^*$ denotes the global optimal solution to (1). $\|x\|_0$ denotes the $\ell_0$ norm of vector $x$, that is, the number of nonzeros in $x$; $e_i \in \mathbb{R}^n$ denotes the $i$th natural unit basis vector. We use $\mathbb{E}_{\xi_{k,*}}(\cdot)$ to denote the expectation with respect to a set of variables $\{\xi_{k,1}, \cdots, \xi_{k,M}\}$. $\mathbb{E}(\cdot)$ means taking the expectation in terms of all random variables. $G(x; \xi)$ is used to denote $\nabla F(x; \xi)$ for short. We use $\nabla_i f(x)$ and $(G(x; \xi))_i$ to denote the $i$th element of $\nabla f(x)$ and $G(x; \xi)$ respectively.

**Assumption** Throughout this paper, we make the following assumption for the objective function. All of them are quite common in the analysis of stochastic gradient algorithms.

**Assumption 1.** *We assume that the following holds:*

- **(Unbiased Gradient):** *The stochastic gradient $G(x; \xi)$ is unbiased, that is to say,*

$$\nabla f(x) = \mathbb{E}_\xi [G(x; \xi)] \tag{2}$$

- **(Bounded Variance):** *The variance of stochastic gradient is bounded:*
$$\mathbb{E}_\xi(\|G(x;\xi) - \nabla f(x)\|^2) \le \sigma^2, \quad \forall x. \tag{3}$$
- **(Lipschitzian Gradient):** *The gradient function $\nabla f(\cdot)$ is Lipschitzian, that is to say,*
$$\|\nabla f(x) - \nabla f(y)\| \le L\|x - y\| \quad \forall x, \forall y. \tag{4}$$
*Under the Lipschitzian gradient assumption, we can define two more constants $L_s$ and $L_{\max}$. Let $s$ be any positive integer. Define $L_s$ to be the minimal constant satisfying the following inequality:*

$$\left\|\nabla f(x) - \nabla f\left(x + \sum_{i \in S} \alpha_i e_i\right)\right\| \le L_s \left\|\sum_{i \in S} \alpha_i e_i\right\|, \quad \forall S \subset \{1, 2, ..., n\} \text{ and } |S| \le s \tag{5}$$
*Define $L_{\max}$ as the minimum constant that satisfies:*
$$|\nabla_i f(x) - \nabla_i f(x + \alpha e_i)| \le L_{\max}|\alpha|, \quad \forall i \in \{1, 2, ..., n\}. \tag{6}$$
*It can be seen that $L_{\max} \le L_s \le L$.*

## 2 Related Work

This section mainly reviews asynchronous parallel gradient algorithms, and asynchronous parallel stochastic gradient algorithms and refer readers to the long version of this paper[2] for review of stochastic gradient algorithms and synchronous parallel stochastic gradient algorithms.

The *asynchronous parallel algorithms* received broad attention in optimization recently, although pioneer studies started from 1980s [Bertsekas and Tsitsiklis, 1989]. Due to the rapid development of hardware resources, the asynchronous parallelism recently received many successes when applied to parallel stochastic gradient [Niu et al., 2011, Agarwal and Duchi, 2011, Zhang et al., 2014, Feyzmahdavian et al., 2015, Paine et al., 2013], stochastic coordinate descent [Avron et al., 2014, Liu et al., 2014a], dual stochastic coordinate ascent [Tran et al., 2015], randomized Kaczmarz algorithm [Liu et al., 2014b], and ADMM [Zhang and Kwok, 2014]. Liu et al. [2014a] and Liu and Wright [2014] studied the asynchronous parallel stochastic coordinate descent algorithm with consistent read and inconsistent read respectively and prove the linear speedup is achievable if $T \le O(n^{1/2})$ for smooth convex functions and $T \le O(n^{1/4})$ for functions with "smooth convex loss + nonsmooth convex separable regularization". Avron et al. [2014] studied this asynchronous parallel stochastic coordinate descent algorithm in solving $Ax = b$ where $A$ is a symmetric positive definite matrix, and showed that the linear speedup is achievable if $T \le O(n)$ for consistent read and $T \le O(n^{1/2})$ for inconsistent read. Tran et al. [2015] studied a semi-asynchronous parallel version of Stochastic Dual Coordinate Ascent algorithm which periodically enforces primal-dual synchronization in a separate thread.

We review the *asynchronous parallel stochastic gradient algorithms* in the last. Agarwal and Duchi [2011] analyzed the ASYSG-CON algorithm (on computer cluster) for *convex smooth* optimization and proved a convergence rate of $O(1/\sqrt{MK} + MT^2/K)$ which implies that linear speedup is achieved when $T$ is bounded by $O(K^{1/4}/M^{3/4})$. In comparison, our analysis for the more general nonconvex smooth optimization improves the upper bound by a factor $K^{1/4}M^{1/4}$. A very recent work [Feyzmahdavian et al., 2015] extended the analysis in Agarwal and Duchi [2011] to minimize functions in the form "smooth convex loss + nonsmooth convex regularization" and obtained similar results. Niu et al. [2011] proposed a lock free asynchronous parallel implementation of SG on the shared memory system and described this implementation as HOGWILD! algorithm. They proved a sublinear convergence rate $O(1/K)$ for *strongly* convex smooth objectives. Another recent work Mania et al. [2015] analyzed asynchronous stochastic optimization algorithms for convex functions by viewing it as a serial algorithm with the input perturbed by bounded noise and proved the convergences rates no worse than using traditional point of view for several algorithms.

## 3 Asynchronous parallel stochastic gradient for computer network

This section considers the asynchronous parallel implementation of SG on computer network proposed by Agarwal and Duchi [2011]. It has been successfully applied to the distributed neural network [Dean et al., 2012] and the parameter server [Li et al., 2014a] to solve deep neural network.

## 3.1 Algorithm Description: ASYSG-CON

The "star" in the star-shaped network is a master machine[3] which maintains the parameter $x$. Other machines in the computer network serve as workers which only communicate with the master. All workers exchange information with the master independently and simultaneously, basically repeating the following steps:

---

**Algorithm 1** ASYSG-CON

**Require:** $x_0, K, \{\gamma_k\}_{k=0,\cdots,K-1}$
**Ensure:** $x_K$
1: **for** $k = 0, \cdots, K-1$ **do**
2:     Randomly select $M$ training samples indexed by $\xi_{k,1}, \xi_{k,2}, ...\xi_{k,M}$;
3:     $x_{k+1} = x_k - \gamma_k \sum_{m=1}^{M} G(x_{k-\tau_{k,m}}, \xi_{k,m})$;
4: **end for**

---

- **(Select):** randomly select a subset of training samples $S \in \Xi$;
- **(Pull):** pull parameter $x$ from the master;
- **(Compute):** compute the stochastic gradient $g \leftarrow \sum_{\xi \in S} G(x; \xi)$;
- **(Push):** push $g$ to the master.

The master basically repeats the following steps:

- **(Aggregate):** aggregate a certain amount of stochastic gradients "$g$" from workers;
- **(Sum):** summarize all "$g$"s into a vector $\Delta$;
- **(Update):** update parameter $x$ by $x \leftarrow x - \gamma\Delta$.

While the master is aggregating stochastic gradients from workers, it does not care about the sources of the collected stochastic gradients. As long as the total amount achieves the predefined quantity, the master will compute $\Delta$ and perform the update on $x$. The "update" step is performed as an atomic operation – workers cannot read the value of $x$ during this step, which can be efficiently implemented in the network (especially in the parameter server [Li et al., 2014a]). The key difference between this asynchronous parallel implementation of SG and the serial (or synchronous parallel) SG algorithm lies on that in the "update" step, some stochastic gradients "$g$" in "$\Delta$" might be computed from some early value of $x$ instead of the current one, while in the serial SG, all $g$'s are guaranteed to use the current value of $x$.

The asynchronous parallel implementation substantially reduces the system overhead and overcomes the possible large network delay, but the cost is to use the old value of "$x$" in the stochastic gradient evaluation. We will show in Section 3.2 that the negative affect of this cost will vanish asymptotically.

To mathematically characterize this asynchronous parallel implementation, we monitor parameter $x$ in the master. We use the subscript $k$ to indicate the $k$th iteration on the master. For example, $x_k$ denotes the value of parameter $x$ after $k$ updates, so on and so forth. We introduce a variable $\tau_{k,m}$ to denote how many delays for $x$ used in evaluating the $m$th stochastic gradient at the $k$th iteration. This asynchronous parallel implementation of SG on the "star-shaped" network is summarized by the ASYSG-CON algorithm, see Algorithm 1. The suffix "CON" is short for "consistent read". "Consistent read" means that the value of $x$ used to compute the stochastic gradient is a real state of $x$ no matter at which time point. "Consistent read" is ensured by the atomicity of the "update" step. When the atomicity fails, it leads to "inconsistent read" which will be discussed in Section 4. It is worth noting that on some "non-star" structures the asynchronous implementation can also be described as ASYSG-CON in Algorithm 1, for example, the cyclic delayed architecture and the locally averaged delayed architecture [Agarwal and Duchi, 2011, Figure 2] .

### 3.2 Analysis for ASYSG-CON

To analyze Algorithm 1, besides Assumption 1 we make the following additional assumptions.

**Assumption 2.** *We assume that the following holds:*

- **(Independence):** *All random variables in $\{\xi_{k,m}\}_{k=0,1,\cdots,K;m=1,\cdots,M}$ in Algorithm 1 are independent to each other;*
- **(Bounded Age):** *All delay variables $\tau_{k,m}$'s are bounded:* $\max_{k,m} \tau_{k,m} \leq T$.

The independence assumption strictly holds if all workers select samples with *replacement*. Although it might not be satisfied strictly in practice, it is a common assumption made for the analysis

purpose. The bounded delay assumption is much more important. As pointed out before, the asynchronous implementation may use some old value of parameter $x$ to evaluate the stochastic gradient. Intuitively, the age (or "oldness") should not be too large to ensure the convergence. Therefore, it is a natural and reasonable idea to assume an upper bound for ages. This assumption is commonly used in the analysis for asynchronous algorithms, for example, [Niu et al., 2011, Avron et al., 2014, Liu and Wright, 2014, Liu et al., 2014a, Feyzmahdavian et al., 2015, Liu et al., 2014b]. It is worth noting that the upper bound $T$ is roughly proportional to the number of workers.

Under Assumptions 1 and 2, we have the following convergence rate for nonconvex optimization.

**Theorem 1.** *Assume that Assumptions 1 and 2 hold and the steplength sequence $\{\gamma_k\}_{k=1,\cdots,K}$ in Algorithm 1 satisfies*

$$LM\gamma_k + 2L^2M^2T\gamma_k \sum_{\kappa=1}^{T} \gamma_{k+\kappa} \leq 1 \quad \textit{for all } k = 1, 2, \dots. \tag{7}$$

*We have the following ergodic convergence rate for the iteration of Algorithm 1*

$$\frac{1}{\sum_{k=1}^{K} \gamma_k} \sum_{k=1}^{K} \gamma_k \mathbb{E}(\|\nabla f(x_k)\|^2) \leq \frac{2(f(x_1)-f(x^*))+\sum_{k=1}^{K}\left(\gamma_k^2 ML + 2L^2M^2\gamma_k \sum_{j=k-T}^{k-1} \gamma_j^2\right)\sigma^2}{M\sum_{k=1}^{K}\gamma_k}. \tag{8}$$

*where $\mathbb{E}(\cdot)$ denotes taking expectation in terms of all random variables in Algorithm 1.*

To evaluate the convergence rate, the commonly used metrics in convex optimization are not eligible, for example, $f(x_k) - f^*$ and $\|x_k - x^*\|^2$. For nonsmooth optimization, we use the ergodic convergence as the metric, that is, the weighted average of the $\ell_2$ norm of all gradients $\|\nabla f(x_k)\|^2$, which is used in the analysis for nonconvex optimization [Ghadimi and Lan, 2013]. Although the metric used in nonconvex optimization is not exactly comparable to $f(x_k) - f^*$ or $\|x_k - x^*\|^2$ used in the analysis for convex optimization, it is not totally unreasonable to think that they are roughly in the same order. The ergodic convergence directly indicates the following convergence: If randomly select an index $\tilde{K}$ from $\{1, 2, \cdots, K\}$ with probability $\{\gamma_k / \sum_{k=1}^{K} \gamma_k\}$, then $\mathbb{E}(\|\nabla f(x_{\tilde{K}})\|^2)$ is bounded by the right hand side of (8) and all bounds we will show in the following.

Taking a close look at Theorem 1, we can properly choose the steplength $\gamma_k$ as a constant value and obtain the following convergence rate:

**Corollary 2.** *Assume that Assumptions 1 and 2 hold. Set the steplength $\gamma_k$ to be a constant $\gamma$*

$$\gamma := \sqrt{f(x_1) - f(x^*)/(MLK\sigma^2)}. \tag{9}$$

*If the delay parameter $T$ is bounded by*

$$K \geq 4ML(f(x_1) - f(x^*))(T+1)^2/\sigma^2, \tag{10}$$

*then the output of Algorithm 1 satisfies the following ergodic convergence rate*

$$\min_{k\in\{1,\cdots,K\}} \mathbb{E}\|\nabla f(x_k)\|^2 \leq \frac{1}{K}\sum_{k=1}^{K} \mathbb{E}\|\nabla f(x_k)\|^2 \leq 4\sqrt{(f(x_1)-f(x^*))L/(MK)}\sigma. \tag{11}$$

This corollary basically claims that when the total iteration number $K$ is greater than $O(T^2)$, the convergence rate achieves $O(1/\sqrt{MK})$. Since this rate does not depend on the delay parameter $T$ after sufficient number of iterations, the negative effect of using old values of $x$ for stochastic gradient evaluation vanishes asymptoticly. In other words, if the total number of workers is bounded by $O(\sqrt{K/M})$, the linear speedup is achieved.

Note that our convergence rate $O(1/\sqrt{MK})$ is consistent with the serial SG (with $M = 1$) for convex optimization [Nemirovski et al., 2009], the synchronous parallel (or mini-batch) SG for convex optimization [Dekel et al., 2012], and nonconvex smooth optimization [Ghadimi and Lan, 2013]. Therefore, an important observation is that as long as the number of workers (which is proportional to $T$) is bounded by $O(\sqrt{K/M})$, the iteration complexity to achieve the same accuracy level will be roughly the same. In other words, the average work load for each worker is reduced by the factor $T$ comparing to the serial SG. Therefore, the linear speedup is achievable if $T \leq O(\sqrt{K/M})$. Since our convergence rate meets several special cases, it is tight.

Next we compare with the analysis of ASYSG-CON for *convex* smooth optimization in Agarwal and Duchi [2011, Corollary 2]. They proved an asymptotic convergence rate $O(1/\sqrt{MK})$, which is consistent with ours. But their results require $T \leq O(K^{1/4}M^{-3/4})$ to guarantee linear speedup. Our result improves it by a factor $O(K^{1/4}M^{1/4})$.

# 4 Asynchronous parallel stochastic gradient for shared memory architecture

This section considers a widely used lock-free asynchronous implementation of SG on the shared memory system proposed in Niu et al. [2011]. Its advantages have been witnessed in solving SVM, graph cuts [Niu et al., 2011], linear equations [Liu et al., 2014b], and matrix completion [Petroni and Querzoni, 2014]. While the computer network always involves multiple machines, the shared memory platform usually only includes a single machine with multiple cores / GPUs sharing the same memory.

## 4.1 Algorithm Description: ASYSG-INCON

For the shared memory platform, one can exactly follow ASYSG-CON on the computer network using software locks, which is expensive[4]. Therefore, in practice the lock free asynchronous parallel implementation of SG is preferred. This section considers the same implementation as Niu et al. [2011], but provides a more precise algorithm description ASYSG-INCON than HOGWILD! proposed in Niu et al. [2011].

---

**Algorithm 2** ASYSG-INCON

**Require:** $x_0, K, \gamma$
**Ensure:** $x_K$
1: **for** $k = 0, \cdots, K - 1$ **do**
2:   Randomly select $M$ training samples indexed by $\xi_{k,1}, \xi_{k,2}, ... \xi_{k,M}$;
3:   Randomly select $i_k \in \{1, 2, ..., n\}$ with uniform distribution;
4:   $(x_{k+1})_{i_k} = (x_k)_{i_k} - \gamma \sum_{m=1}^{M} (G(\hat{x}_{k,m}; \xi_{k,m}))_{i_k}$;
5: **end for**

---

In this lock free implementation, the shared memory stores the parameter "$x$" and allows all workers reading and modifying parameter $x$ simultaneously without using locks. All workers repeat the following steps independently, concurrently, and simultaneously:

- **(Read):** read the parameter from the shared memory to the local memory *without software locks* (we use $\hat{x}$ to denote its value);
- **(Compute):** sample a training data $\xi$ and use $\hat{x}$ to compute the stochastic gradient $G(\hat{x}; \xi)$ *locally*;
- **(Update):** update parameter $x$ in the shared memory *without software locks* $x \leftarrow x - \gamma G(\hat{x}; \xi)$.

Since we do not use locks in both "read" and "update" steps, it means that multiple workers may manipulate the shared memory simultaneously. It causes the "inconsistent read" at the "read" step, that is, the value of $\hat{x}$ read from the shared memory might not be any state of $x$ in the shared memory at any time point. For example, at time 0, the original value of $x$ in the shared memory is a two dimensional vector $[a, b]$; at time 1, worker $W$ is running the "read" step and first reads $a$ from the shared memory; at time 2, worker $W'$ updates the first component of $x$ in the shared memory from $a$ to $a'$; at time 2, worker $W'$ updates the second component of $x$ in the shared memory from $b$ to $b'$; at time 3, worker $W$ reads the value of the second component of $x$ in the shared memory as $b'$. In this case, worker $W$ eventually obtains the value of $\hat{x}$ as $[a, b']$, which is not a real state of $x$ in the shared memory at any time point. Recall that in ASYSG-CON the parameter value obtained by any worker is guaranteed to be some real value of parameter $x$ at some time point.

To precisely characterize this implementation and especially represent $\hat{x}$, we monitor the value of parameter $x$ in the shared memory. We define one *iteration* as a modification on any *single* component of $x$ in the shared memory since the update on a single component can be considered to be atomic on GPUs and DSPs [Niu et al., 2011]. We use $x_k$ to denote the value of parameter $x$ in the shared memory after $k$ iterations and $\hat{x}_k$ to denote the value read from the shared memory and used for computing stochastic gradient at the $k$th iteration. $\hat{x}_k$ can be represented by $x_k$ with a few earlier updates missing

$$\hat{x}_k = x_k - \sum_{j \in J(k)} (x_{j+1} - x_j) \tag{12}$$

where $J(k) \subset \{k - 1, k, \cdots, 0\}$ is a subset of index numbers of previous iterations. This way is also used in analyzing asynchronous parallel coordinate descent algorithms in [Avron et al., 2014, Liu and Wright, 2014]. The $k$th update happened in the shared memory can be described as

$$(x_{k+1})_{i_k} = (x_k)_{i_k} - \gamma (G(\hat{x}_k; \xi_k))_{i_k}$$

where $\xi_k$ denotes the index of the selected data and $i_k$ denotes the index of the component being updated at $k$th iteration. In the original analysis for the HOGWILD! implementation [Niu et al., 2011], $\hat{x}_k$ is assumed to be some earlier state of $x$ in the shared memory (that is, the consistent read) for simpler analysis, although it is not true in practice.

One more complication is to apply the mini-batch strategy like before. Since the "update" step needs physical modification in the shared memory, it is usually much more time consuming than both "read" and "compute" steps are. If many workers run the "update" step simultaneously, the memory contention will seriously harm the performance. To reduce the risk of memory contention, a common trick is to ask each worker to gather multiple (say $M$) stochastic gradients and write the shared memory only once. That is, in each cycle, run both "update" and "compute" steps for $M$ times before you run the "update" step. Thus, the mini-batch updates happen in the shared memory can be written as

$$(x_{k+1})_{i_k} = (x_k)_{i_k} - \gamma \sum_{m=1}^{M} (G(\hat{x}_{k,m}; \xi_{k,m}))_{i_k} \qquad (13)$$

where $i_k$ denotes the coordinate index updated at the $k$th iteration, and $G(\hat{x}_{k,m}; \xi_{k,m})$ is the $m$th stochastic gradient computed from the data sample indexed by $\xi_{k,m}$ and the parameter value denoted by $\hat{x}_{k,m}$ at the $k$th iteration. $\hat{x}_{k,m}$ can be expressed by:

$$\hat{x}_{k,m} = x_k - \sum_{j \in J(k,m)} (x_{j+1} - x_j) \qquad (14)$$

where $J(k,m) \subset \{k-1, k, \cdots, 0\}$ is a subset of index numbers of previous iterations. The algorithm is summarized in Algorithm 2 from the view of the shared memory.

## 4.2 Analysis for ASYSG-INCON

To analyze the ASYSG-INCON, we need to make a few assumptions similar to Niu et al. [2011], Liu et al. [2014b], Avron et al. [2014], Liu and Wright [2014].

**Assumption 3.** *We assume that the following holds for Algorithm 2:*

- **(Independence):** *All groups of variables $\{i_k, \{\xi_{k,m}\}_{m=1}^{M}\}$ at different iterations from $k = 1$ to $K$ are independent to each other.*
- **(Bounded Age):** *Let $T$ be the global bound for delay: $J(k,m) \subset \{k-1, ...k-T\}, \quad \forall k, \forall m,$ so $|J(k,m)| \leq T$.*

The independence assumption might not be true in practice, but it is probably the best assumption one can make in order to analyze the asynchronous parallel SG algorithm. This assumption was also used in the analysis for HOGWILD! [Niu et al., 2011] and asynchronous randomized Kaczmarz algorithm [Liu et al., 2014b]. The bounded delay assumption basically restricts the age of all missing components in $\hat{x}_{k,m}$ ($\forall m, \forall k$). The upper bound "$T$" here serves a similar purpose as in Assumption 2. Thus we abuse this notation in this section. The value of $T$ is proportional to the number of workers and does not depend on the size of mini-batch $M$. The bounded age assumption is used in the analysis for asynchronous stochastic coordinate descent with "inconsistent read" [Avron et al., 2014, Liu and Wright, 2014]. Under Assumptions 1 and 3, we have the following results:

**Theorem 3.** *Assume that Assumptions 1 and 3 hold and the constant steplength $\gamma$ satisfies*

$$2M^2 T L_T^2 (\sqrt{n} + T - 1)\gamma^2 / n^{3/2} + 2M L_{\max} \gamma \leq 1. \qquad (15)$$

*We have the following ergodic convergence rate for Algorithm 2*

$$\frac{1}{K} \sum_{t=1}^{K} \mathbb{E}\left(\|\nabla f(x_t)\|^2\right) \leq \frac{2n}{KM\gamma}(f(x_1) - f(x^*)) + \frac{L_T^2 TM\gamma^2}{2n}\sigma^2 + L_{\max}\gamma\sigma^2. \qquad (16)$$

Taking a close look at Theorem 3, we can choose the steplength $\gamma$ properly and obtain the following error bound:

**Corollary 4.** *Assume that Assumptions 1 and 3 hold. Set the steplength to be a constant $\gamma$*

$$\gamma := \sqrt{2(f(x_1) - f(x^*))n} / (\sqrt{K L_T M}\sigma). \qquad (17)$$

*If the total iterations $K$ is greater than*

$$K \geq 16(f(x_1) - f(x^*))L_T M \left(n^{3/2} + 4T^2\right) / (\sqrt{n}\sigma^2), \qquad (18)$$

*then the output of Algorithm 2 satisfies the following ergodic convergence rate*

$$\frac{1}{K} \sum_{k=1}^{K} \mathbb{E}(\|\nabla f(x_k)\|^2) \leq \sqrt{72 \left(f(x_1) - f(x^*)\right) L_T n / (KM)}\sigma. \qquad (19)$$

This corollary indicates the asymptotic convergence rate achieves $O(1/\sqrt{MK})$ when the total iteration number $K$ exceeds a threshold in the order of $O(T^2)$ (if $n$ is considered as a constant). We can see that this rate and the threshold are consistent with the result in Corollary 2 for ASYSG-CON. One may argue that why there is an additional factor $\sqrt{n}$ in the numerator of (19). That is due to the way we count iterations – one iteration is defined as updating a single component of $x$. If we take into account this factor in the comparison to ASYSG-CON, the convergence rates for ASYSG-CON and ASYSG-INCON are essentially consistent. This comparison implies that the "inconsistent read" would not make a big difference from the "consistent read".

Next we compare our result with the analysis of HOGWILD! by [Niu et al., 2011]. In principle, our analysis and their analysis consider the same implementation of asynchronous parallel SG, but differ in the following aspects: 1) our analysis considers the smooth nonconvex optimization which includes the smooth strongly convex optimization considered in their analysis; 2) our analysis considers the "inconsistent read" model which meets the practice while their analysis assumes the impractical "consistent read" model. Although the two results are not absolutely comparable, it is still interesting to see the difference. Niu et al. [2011] proved that the linear speedup is achievable if the maximal number of nonzeros in stochastic gradients is bounded by $O(1)$ and the number of workers is bounded by $O(n^{1/4})$. Our analysis does not need this prerequisite and guarantees the linear speedup as long as the number of workers is bounded by $O(\sqrt{K})$. Although it is hard to say that our result strictly dominates HOGWILD! in Niu et al. [2011], our asymptotic result is eligible for more scenarios.

## 5  Experiments

The successes of ASYSG-CON and ASYSG-INCON and their advantages over synchronous parallel algorithms have been widely witnessed in many applications such as deep neural network [Dean et al., 2012, Paine et al., 2013, Zhang et al., 2014, Li et al., 2014a], matrix completion [Niu et al., 2011, Petroni and Querzoni, 2014, Yun et al., 2013], SVM [Niu et al., 2011], and linear equations [Liu et al., 2014b]. We refer readers to these literatures for more comphrehensive comparison and empirical studies. This section mainly provides the empirical study to validate the speedup properties for *completeness*. Due to the space limit, please find it in Supplemental Materials.

## 6  Conclusion

This paper studied two popular asynchronous parallel implementations for $SG$ on computer cluster and shared memory system respectively. Two algorithms (ASYSG-CON and ASYSG-INCON) are used to describe two implementations. An asymptotic sublinear convergence rate is proven for both algorithms on *nonconvex* smooth optimization. This rate is consistent with the result of $SG$ for convex optimization. The linear speedup is proven to achievable when the number of workers is bounded by $\sqrt{K}$, which improves the earlier analysis of ASYSG-CON for convex optimization in [Agarwal and Duchi, 2011]. The proposed ASYSG-INCON algorithm provides a more precise description for lock free implementation on shared memory system than HOGWILD! [Niu et al., 2011]. Our result for ASYSG-INCON can be applied to more scenarios.

**Acknowledgements**

This project is supported by the NSF grant CNS-1548078, the NEC fellowship, and the startup funding at University of Rochester. We thank Professor Daniel Gildea and Professor Sandhya Dwarkadas at University of Rochester, Professor Stephen J. Wright at University of Wisconsin-Madison, and anonymous (meta-)reviewers for their constructive comments and helpful advices.

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

## Footnotes

[1]The speedup for $T$ workers is defined as the ratio between the total work load using one worker and the average work load using $T$ workers to obtain a solution at the same precision. "The linear speedup is achieved" means that the speedup with $T$ workers greater than $cT$ for any values of $T$ ($c \in (0, 1]$ is a constant independent to $T$).

[2]http://arxiv.org/abs/1506.08272

[3]There could be more than one machines in some networks, but all of them serves the same purpose and can be treated as a single machine.

[4]The time consumed by locks is roughly equal to the time of $10^4$ floating-point computation. The additional cost for using locks is the waiting time during which multiple worker access the same memory address.
