[Supplementary Material · nips2015-supp.pdf]

# Supplemental Materials for Asynchronous Parallel Stochastic Gradient for Nonconvex Optimization: Experiments and Proofs

## A   Experiments

The successes of ASYSG-CON and ASYSG-INCON and their advantages over synchronous parallel algorithms have been widely witnessed in many applications such as deep neural network [Dean et al., 2012, Paine et al., 2013, Zhang et al., 2014, Li et al., 2014a], matrix completion [Niu et al., 2011, Petroni and Querzoni, 2014, Yun et al., 2013], SVM [Niu et al., 2011], and solving linear equations [Liu et al., 2014b]. We refer readers to these literatures for more comphrehensive comparison and empirical studies. This section mainly provides the empirical study to validate the speedup properties for *completeness*.

We perform experiments for ASYSG-CON and ASYSG-INCON on computer cluster and multicore machine respectively. The main purpose of the following experiments is to validate the speedup property. We are particularly interested in two types of speedup: iteration speedup and running time speedup. The iteration speedup is exactly the speedup we discussed in the whole paper. Given $T$ workers, it is computed from the ratio

$$\text{iteration speedup of } T \text{ workers} = \frac{\text{\# of total iterations of the serial SG (or using one worker)}}{\text{\# of total iterations using } T \text{ workers}} \times T$$

where # is the iteration count when the same level of precision achieved. This speedup is less affected by the hardware. The running time speedup is the actual speedup. It is defined with respect to the running time:

$$\text{running time speedup of } T \text{ workers} = \frac{\text{running time for the serial SG (or using one worker)}}{\text{running time of using } T \text{ workers}}.$$

The running time speedup is seriously affected by the hardware. It is generally worse than the iteration speedup.

### A.1   ASYSG-CON

We implement ASYSG-CON for deep neural network based on the Caffe [Jia et al., 2014] package. Caffe is an open source code base of deep learning algorithms. We evaluate ASYSG-CON on two standard datasets provided in the Caffe package, LENET and CIFAR10-FULL.

The neural network consists of convolution layer, nonlinear layer, max pool layer and fully connected layer [Krizhevsky et al., 2012], and the detailed specification can be found on the Caffe website[5]. LENET is a digit classifier network, training on the MNIST dataset[6]. CIFAR10-full has 10 classes of color images, training on the CIFAR10 dataset [Krizhevsky and Hinton, 2009]. We first initialize a parameter server hosting the parameters, and then spawn up to 8 stochastic gradient workers. The point to point communication between the parameter server and gradient workers are handled by the MPICH library[7]. The parameter server and stochastic gradient workers run on separate machines, and each process uses a single core of a Xeon(R) E5-2430 CPU. The steplength $\gamma$ for LENET is chosen as the default value. It means that this steplength has been tuned to be the optimal for the serial SG algorithm on LENET. The steplength we used for CIFAR10-FULL is chosen as the default value as well.

We draw the curves of objective loss against iterations and running time in Figures 1 and 2 respectively, and report their speedups in Tables 1 and 2. (More details about datasets and the parameter setting in experiments can be found in Table 3.) We can observe that

- The iteration speedup is always better than the running time speedup, which meets our common sense;
- The speedups for both problems are comparable overall as shown in Tables 1 and 2. The time speedup for CIFAR10-FULL is slightly more stable, while we notice that the time speedup for LENET in Table 1 suddenly drops to "2.88" with mpi-8 from "5.29" with mpi-7. That is because it hits the ceiling of communication bandwidth. The number of parameters in LENET is more than CIFAR10-FULL, thus requiring more communication cost. When the number of machines achieves a certain threshold (in this case LENET, it is 8), the performance might become dramatically worse.

### A.2   ASYSG-INCON

We conduct the empirical study for ASYSG-INCON on the machine (Intel Xeon architecture), which has 4 sockets and 10 cores for each socket. The synthetic data is generated from a full connected neural network with

Figure 1: LENET. The ASYSG-CON algorithm is run on various numbers of machines from 1 to 8 to solve LENET. The curves of the objective loss against the number of iteration and the running time are drawn in the left and the right graphs respectively.

Figure 2: CIFAR10-FULL. The ASYSG-CON algorithm is run on various numbers of machines from 1 to 8 to solve CIFAR10-FULL. The curves of the objective loss against the number of iteration and the running time are drawn in the left and the right graphs respectively.

Table 1: Iteration speedup and running time speedup of ASYSG-CON. (LENET)

|  | mpi-1 | mpi-2 | mpi-3 | mpi-4 | mpi-5 | mpi-6 | mpi-7 | mpi-8 |
|---|---|---|---|---|---|---|---|---|
| iteration speedup | 1.07 | 2.02 | 2.77 | 3.73 | 4.20 | 5.82 | 6.86 | 6.97 |
| time speedup | 0.98 | 1.69 | 2.24 | 2.96 | 3.30 | 4.51 | 5.29 | 2.88 |

Table 2: Iteration speedup and running time speedup of ASYSG-CON. (CIFAR10-FULL)

|  | mpi-1 | mpi-2 | mpi-3 | mpi-4 | mpi-5 | mpi-6 | mpi-7 | mpi-8 |
|---|---|---|---|---|---|---|---|---|
| iteration speedup | 1.01 | 1.93 | 2.65 | 3.42 | 4.27 | 4.92 | 5.36 | 5.96 |
| time speedup | 1.00 | 1.73 | 2.28 | 2.88 | 3.56 | 4.07 | 4.41 | 5.00 |

Table 3: More details about LENET and CIFAR10-FULL.

| Datasets | Type | #Images | MiniBatch | #CONV | #FC | #Params |
|---|---|---|---|---|---|---|
| LENET | 28x28 grayscale | 60K | 60 | 2 | 2 | 431,080 |
| CIFAR10-FULL | 32x32 RGB | 50K | 100 | 3 | 1 | 89,578 |

5 layers ($400 \times 100 \times 50 \times 20 \times 10$) and $46380$ parameters totally. The total number of samples is $463800$. The data size is about $1.5$ GB. The input vector and all parameters are generated from i.i.d. Gaussian distribution. The output vector is constructed by applying the network parameter to the input vector plus some Gaussian random noise.

We run ASYSG-INCON on various numbers of cores from 1 to 32. The size of mini-batch is chosen as $M = 32$ and the steplength is chosen as $\gamma = 1.1 \times 10^{-7}$. Both parameters are chosen based on the best performance of the serial SG to achieve the precision $10^{-1}$ for the $\ell_2$ norm of gradient. Figure 3 draws the curve of the $\ell_2$ norm of gradients against the number of iterations and running time respectively. The speedup is reported in Table 4. We observe that the iteration speedup is almost linear while the running time speedup is slightly worse than the iteration speedup. The overall performance of ASYSG-INCON on the shared memory system is better than ASYSG-CON on the computer cluster. The reason is that the computer cluster suffers from serious communication delay and the delay bound $T$ on the computer cluster is usually larger than on the shared memory system.

Figure 3: Deep neural network for synthetic data using ASYSG-INCON. The ASYSG-CON algorithm is run on various numbers of machines from 1 to 32. The curves of the objective loss against the number of iteration and the running time are drawn in the left and the right graphs respectively.

Table 4: Iteration speedup and running time speedup of ASYSG-INCON (synthetic data).

|  | thr-1 | thr-4 | thr-8 | thr-12 | thr-16 | thr-20 | thr-24 | thr-28 | thr-32 |
|---|---|---|---|---|---|---|---|---|---|
| iteration speedup | 1 | 3.9 | 7.8 | 11.6 | 15.4 | 19.9 | 24.1 | 28.7 | 31.6 |
| time speedup | 1 | 4.0 | 8.1 | 11.9 | 16.3 | 19.2 | 22.7 | 26.1 | 29.2 |

# B  Proofs

**Proofs to Theorem 1**

*Proof.* From the Lipschitzisan gradient assumption (5), we have

$$
f(x_{k+1}) - f(x_k) \leq \langle \nabla f(x_k), x_{k+1} - x_k \rangle + \frac{L}{2}\|x_{k+1} - x_k\|^2
$$
$$
= -\left\langle \nabla f(x_k), \gamma_k \sum_{m=1}^{M} G(x_{k-\tau_{k,m}}; \xi_{k,m}) \right\rangle + \frac{\gamma_k^2 L}{2}\left\| \sum_{m=1}^{M} G(x_{k-\tau_{k,m}}; \xi_{k,m}) \right\|^2. \quad (20)
$$

Taking expectation respect to $\xi_{k,*}$ on both sides of (20), we have

$$
\mathbb{E}_{\xi_{k,*}}(f(x_{k+1})) - f(x_k) \leq -M\gamma_k \left\langle \nabla f(x_k), \frac{1}{M}\sum_{m=1}^{M}\nabla f(x_{k-\tau_{k,m}}) \right\rangle
$$

$$+ \frac{\gamma_k^2 L}{2} \mathbb{E}_{\xi_k,*} \left( \left\| \sum_{m=1}^{M} G(x_{k-\tau_{k,m}}; \xi_{k,m}) \right\|^2 \right) \tag{21}$$

where we use the unbiased stochastic gradient assumption in (2). From the fact

$$\langle a, b \rangle = \frac{1}{2} \left( \|a\|^2 + \|b\|^2 - \|a - b\|^2 \right),$$

we have

$$\mathbb{E}_{\xi_k,*}(f(x_{k+1})) - f(x_k)$$

$$\leq - \frac{M\gamma_k}{2} \left( \|\nabla f(x_k)\|^2 + \left\| \frac{1}{M} \sum_{m=1}^{M} \nabla f(x_{k-\tau_{k,m}}) \right\|^2 - \underbrace{\left\| \nabla f(x_k) - \frac{1}{M} \sum_{m=1}^{M} \nabla f(x_{k-\tau_{k,m}}) \right\|^2}_{T_1} \right)$$

$$+ \frac{\gamma_k^2 L}{2} \underbrace{\mathbb{E}_{\xi_k,*} \left( \left\| \sum_{m=1}^{M} G(x_{k-\tau_{k,m}}; \xi_{k,m}) \right\|^2 \right)}_{T_2}. \tag{22}$$

Next we estimate the upper bound of $T_1$ and $T_2$. For $T_2$ we have

$$T_2 = \mathbb{E}_{\xi_k,*} \left[ \left\| \sum_{m=1}^{M} G\left( x_{k-\tau_{k,m}}; \xi_{k,m} \right) \right\|^2 \right]$$

$$= \mathbb{E}_{\xi_k,*} \left[ \left\| \sum_{m=1}^{M} \left( G(x_{k-\tau_{k,m}}; \xi_{k,m}) - \nabla f(x_{k-\tau_{k,m}}) \right) + \sum_{m=1}^{M} \nabla f(x_{k-\tau_{k,m}}) \right\|^2 \right]$$

$$= \mathbb{E}_{\xi_k,*} \left[ \left\| \sum_{m=1}^{M} \left( G(x_{k-\tau_{k,m}}; \xi_{k,m}) - \nabla f(x_{k-\tau_{k,m}}) \right) \right\|^2 \right.$$

$$\left. + \left\| \sum_{m=1}^{M} \nabla f(x_{k-\tau_{k,m}}) \right\|^2 + 2 \left\langle \sum_{m=1}^{M} \left( G(x_{k-\tau_{k,m}}; \xi_{k,m}) - \nabla f(x_{k-\tau_{k,m}}) \right), \sum_{m=1}^{M} \nabla f(x_{k-\tau_{k,m}}) \right\rangle \right]$$

$$= \mathbb{E}_{\xi_k,*} \left[ \left\| \sum_{m=1}^{M} \left( G(x_{k-\tau_{k,m}}; \xi_{k,m}) - \nabla f(x_{k-\tau_{k,m}}) \right) \right\|^2 + \left\| \sum_{m=1}^{M} \nabla f(x_{k-\tau_{k,m}}) \right\|^2 \right]$$

$$= \mathbb{E}_{\xi_k,*} \left[ \sum_{m=1}^{M} \left\| \left( G(x_{k-\tau_{k,m}}; \xi_{k,m}) - \nabla f(x_{k-\tau_{k,m}}) \right) \right\|^2 \right.$$

$$+ 2 \sum_{1 \leq m < m' \leq M} \left\langle G(x_{k-\tau_{k,m}}; \xi_{k,m}) - \nabla f(x_{k-\tau_{k,m}}), G(x_{k-\tau_{k,m'}}; \xi_{k,m'}) - \nabla f(x_{k-\tau_{k,m'}}) \right\rangle$$

$$\left. + \left\| \sum_{m=1}^{M} \nabla f(x_{k-\tau_{k,m}}) \right\|^2 \right]$$

$$\leq M\sigma^2 + \left\| \sum_{m=1}^{M} \nabla f(x_{k-\tau_{k,m}}) \right\|^2 \tag{23}$$

where the forth equality is due to

$$\mathbb{E}_{\xi_k,*} \left\langle \sum_{m=1}^{M} \left( G(x_{k-\tau_{k,m}}; \xi_{k,m}) - \nabla f(x_{k-\tau_{k,m}}) \right), \sum_{m=1}^{M} \nabla f(x_{k-\tau_{k,m}}) \right\rangle$$

$$= \left\langle \sum_{m=1}^{M} \mathbb{E}_{\xi_k,*} \left( G(x_{k-\tau_{k,m}}; \xi_{k,m}) - \nabla f(x_{k-\tau_{k,m}}) \right), \sum_{m=1}^{M} \nabla f(x_{k-\tau_{k,m}}) \right\rangle$$

$$= 0$$

and the last inequality is due to the assumption (3) and

$$\mathbb{E}_{\xi_{k,*}} \sum_{1 \le m < m' \le M} \left\langle G(x_{k-\tau_{k,m}}; \xi_{k,m}) - \nabla f(x_{k-\tau_{k,m}}), G(x_{k-\tau_{k,m'}}; \xi_{k,m'}) - \nabla f(x_{k-\tau_{k,m'}}) \right\rangle$$

$$=\mathbb{E}_{\xi_{k,*}} \sum_{1 \le m < m' \le M} \mathbb{E}_{k,m'} \left\langle G(x_{k-\tau_{k,m}}; \xi_{k,m}) - \nabla f(x_{k-\tau_{k,m}}), G(x_{k-\tau_{k,m'}}; \xi_{k,m'}) - \nabla f(x_{k-\tau_{k,m'}}) \right\rangle$$

$$=\mathbb{E}_{\xi_{k,*}} \sum_{1 \le m < m' \le M} \left\langle G(x_{k-\tau_{k,m}}; \xi_{k,m}) - \nabla f(x_{k-\tau_{k,m}}), \mathbb{E}_{k,m'} G(x_{k-\tau_{k,m'}}; \xi_{k,m'}) - \nabla f(x_{k-\tau_{k,m'}}) \right\rangle$$

$$=0. \tag{24}$$

We next turn to $T_1$:

$$T_1 = \left\| \nabla f(x_k) - \frac{1}{M} \sum_{m=1}^{M} \nabla f\left(x_{k-\tau_{k,m}}\right) \right\|^2$$

$$=\frac{1}{M^2} \left\| \sum_{m=1}^{M} \left(\nabla f(x_k) - \nabla f\left(x_{k-\tau_{k,m}}\right)\right) \right\|^2$$

$$\le \frac{1}{M} \sum_{m=1}^{M} \left\| \left(\nabla f(x_k) - \nabla f\left(x_{k-\tau_{k,m}}\right)\right) \right\|^2$$

$$\le \frac{L^2}{M} \sum_{m=1}^{M} \left\| x_k - x_{k-\tau_{k,m}} \right\|^2$$

$$\le L^2 \max_{k \in \{1,\cdots,M\}} \left\| x_k - x_{k-\tau_{k,m}} \right\|^2$$

$$= L^2 \left\| x_k - x_{k-\tau_{k,\mu}} \right\|^2. \quad (\text{let } \mu := \arg\max_{m \in \{1,\cdots,M\}} \left\| x_k - x_{k-\tau_{k,m}} \right\|^2)$$

where the second inequality is from the Lipschitzian gradient assumption (5). It follows that

$$T_1 \le L^2 \left\| x_k - x_{k-\tau_{k,\mu}} \right\|^2$$

$$=L^2 \left\| \sum_{j=k-\tau_{k,\mu}}^{k-1} (x_{j+1} - x_j) \right\|^2$$

$$=L^2 \left\| \sum_{j=k-\tau_{k,\mu}}^{k-1} \gamma_j \sum_{m=1}^{M} G\left(x_{j-\tau_{j,m}}; \xi_{j,m}\right) \right\|^2$$

$$=L^2 \left\| \sum_{j=k-\tau_{k,\mu}}^{k-1} \gamma_j \sum_{m=1}^{M} \left[ G\left(x_{j-\tau_{j,m}}; \xi_{j,m}\right) - \nabla f\left(x_{j-\tau_{j,m}}\right)\right] + \sum_{j=k-\tau_{k,\mu}}^{k-1} \gamma_j \sum_{m=1}^{M} \nabla f\left(x_{j-\tau_{j,m}}\right) \right\|^2$$

$$\le 2L^2 \left( \underbrace{\left\| \sum_{j=k-\tau_{k,\mu}}^{k-1} \gamma_j \sum_{m=1}^{M} \left[ G\left(x_{j-\tau_{j,m}}; \xi_{j,m}\right) - \nabla f\left(x_{j-\tau_{j,m}}\right)\right] \right\|^2}_{T_3} + \underbrace{\left\| \sum_{j=k-\tau_{k,\mu}}^{k-1} \gamma_j \sum_{m=1}^{M} \nabla f\left(x_{j-\tau_{j,m}}\right) \right\|^2}_{T_4} \right)$$

$$\tag{25}$$

where the last inequality uses the fact that $\|a + b\|^2 \le 2\|a\|^2 + 2\|b\|^2$ for any real vectors $a$ and $b$. Taking the expectation in terms of $\{\xi_{j,*} | j \in \{k - \tau_{k,\mu}, ..., k - 1\}\}$ for $T_3$, we have

$$\mathbb{E}_{\xi_{j,*}, j \in \{k-\tau_{k,\mu},...,k-1\}} (T_3)$$

$$=\mathbb{E}_{\xi_{j,*}, j \in \{k-\tau_{k,\mu},...,k-1\}} \left( \left\| \sum_{j=k-\tau_{k,\mu}}^{k-1} \gamma_j \sum_{m=1}^{M} \left[ G(x_{j-\tau_{j,m}}; \xi_{j,m}) - \nabla f(x_{j-\tau_{j,m}})\right] \right\|^2 \right)$$

$$=\mathbb{E}_{\xi_{j,*}, j \in \{k-\tau_{k,\mu},...,k-1\}} \left( \sum_{j=k-\tau_{k,\mu}}^{k-1} \gamma_j^2 \left\| \sum_{m=1}^{M} \left[ G(x_{j-\tau_{j,m}}; \xi_{j,m}) - \nabla f(x_{j-\tau_{j,m}})\right] \right\|^2 \right)$$

$$+ 2\mathbb{E}_{\xi_{j,*}, j \in \{k-\tau_{k,\mu},\ldots,k-1\}} \left( \sum_{k-1 \geq j'' > j' \geq k-\tau_{k,\mu}} \gamma_{j'} \gamma_{j''} \left\langle \sum_{m=1}^{M} \left[ G(x_{j''-\tau_{j'',m}}; \xi_{j'',m}) - \nabla f(x_{j''-\tau_{j'',m}}) \right], \right.\right.$$

$$\left.\left. \sum_{m=1}^{M} \left[ G(x_{j'-\tau_{j',m}}; \xi_{j',m}) - \nabla_{i_{j'}} f(x_{j'-\tau_{j',m}}) \right] \right\rangle \right)$$

$$= \mathbb{E}_{\xi_{j,*}, j \in \{k-\tau_{k,\mu},\ldots,k-1\}} \left( \sum_{j=k-\tau_{k,\mu}}^{k-1} \gamma_j^2 \left\| \sum_{m=1}^{M} \left[ G(x_{j-\tau_{j,m}}; \xi_{j,m}) - \nabla f(x_{j-\tau_{j,m}}) \right] \right\|^2 \right)$$

$$= \mathbb{E}_{\xi_{j,*}, j \in \{k-\tau_{k,\mu},\ldots,k-1\}} \left( \sum_{j=k-\tau_{k,\mu}}^{k-1} \gamma_j^2 \sum_{m=1}^{M} \left\| \left[ G(x_{j-\tau_{j,m}}; \xi_{j,m}) - \nabla f(x_{j-\tau_{j,m}}) \right] \right\|^2 \right)$$

$$\leq M \sum_{j=k-T}^{k-1} \gamma_j^2 \sigma^2 \tag{26}$$

where the second last equality is due to (24) and the third equality is due to

$$\mathbb{E}_{\xi_j, k-1 \geq j \geq k-\tau_{k,\mu}} \left( \sum_{k-1 \geq j'' > j' \geq k-\tau_{k,\mu}} \gamma_{j'} \gamma_{j''} \left\langle \sum_{m=1}^{M} \left[ G(x_{j''-\tau_{j'',m}}; \xi_{j'',m}) - \nabla f(x_{j''-\tau_{j'',m}}) \right], \right.\right.$$

$$\left.\left. \sum_{m=1}^{M} \left[ G(x_{j'-\tau_{j',m}}; \xi_{j',m}) - \nabla f(x_{j'-\tau_{j',m}}) \right] \right\rangle \right)$$

$$= \mathbb{E}_{\xi_j, k-1 \geq j \geq k-\tau_{k,\mu}} \left( \sum_{k-1 \geq j'' > j' \geq k-\tau_{k,\mu}} \gamma_{j'} \gamma_{j''} \left\langle \sum_{m=1}^{M} \left[ \mathbb{E}_{j''*} G(x_{j''-\tau_{j'',m}}; \xi_{j'',m}) - \nabla f(x_{j''-\tau_{j'',m}}) \right], \right.\right.$$

$$\left.\left. \sum_{m=1}^{M} \left[ G(x_{j'-\tau_{j',m}}; \xi_{j',m}) - \nabla f(x_{j'-\tau_{j',m}}) \right] \right\rangle \right)$$

$$= 0.$$

Taking the expectation in terms of $\xi_{j,*}$ for $T_4$, we have

$$\mathbb{E}_{\xi_{j,*}, j \in \{k-\tau_{k,\mu},\ldots,k-1\}} (T_4)$$

$$= \mathbb{E}_{\xi_{j,*}, j \in \{k-\tau_{k,\mu},\ldots,k-1\}} \left( \left\| \sum_{j=k-\tau_{k,\mu}}^{k-1} \gamma_j \sum_{m=1}^{M} \nabla f\left(x_{j-\tau_{j,m}}\right) \right\|^2 \right)$$

$$\leq T \sum_{j=k-\tau_{k,\mu}}^{k-1} \gamma_j^2 \mathbb{E}_{\xi_{j,*}, j \in \{k-\tau_{k,\mu},\ldots,k-1\}} \left( \left\| \sum_{m=1}^{M} \nabla f(x_{j-\tau_{j,m}}) \right\|^2 \right) \tag{27}$$

where the last inequality uses the upper bound of the delay age: $\tau_{k,\mu} \leq T$.

We take full expectation on both sides of (25) and substitute $\mathbb{E}(T_3)$ and $\mathbb{E}(T_4)$ by their upper bounds in (26) and (27) respectively:

$$\mathbb{E}(T_1) \leq 2L^2 \left( M \sum_{j=k-T}^{k-1} \gamma_j^2 \sigma^2 + T \sum_{j=k-\tau_{k,\mu}}^{k-1} \gamma_j^2 \mathbb{E} \left( \left\| \sum_{m=1}^{M} \nabla f(x_{j-\tau_{j,m}}) \right\|^2 \right) \right). \tag{28}$$

Applying the upper bounds for $\mathbb{E}(T_1)$ in (28) and $\mathbb{E}(T_2)$ in (23) to (22), and take full expectation on both sides, we obtain

$$\mathbb{E}(f(x_{k+1})) - f(x_k)$$

$$\leq -\frac{M\gamma_k}{2} \left[ \mathbb{E}\left( \|\nabla f(x_k)\|^2 \right) + \mathbb{E}\left( \left\| \frac{1}{M} \sum_{m=1}^{M} \nabla f(x_{k-\tau_{k,m}}) \right\|^2 \right) \right.$$

$$\left. - 2L^2 \left( M \sum_{j=k-T}^{k-1} \gamma_j^2 \sigma^2 + T \sum_{j=k-\tau_{k,\mu}}^{k-1} \gamma_j^2 \mathbb{E} \left( \left\| \sum_{m=1}^{M} \nabla f(x_{j-\tau_{j,m}}) \right\|^2 \right) \right) \right]$$

$$+ \frac{\gamma_k^2 L}{2}\left(M\sigma^2 + \mathbb{E}\left(\left\|\sum_{m=1}^M \nabla f(x_{k-\tau_{k,m}})\right\|^2\right)\right)$$

$$\leq -\frac{M\gamma_k}{2}\mathbb{E}\|\nabla f(x_k)\|^2 + \left(\frac{\gamma_k^2 L}{2} - \frac{\gamma_k}{2M}\right)\mathbb{E}\left(\left\|\sum_{m=1}^M \nabla f(x_{k-\tau_{k,m}})\right\|^2\right)$$

$$+ \left(\frac{\gamma_k^2 ML}{2} + L^2 M^2 \gamma_k \sum_{j=k-T}^{k-1} \gamma_j^2\right)\sigma^2$$

$$+ L^2 MT\gamma_k \sum_{j=k-T}^{k-1} \gamma_j^2 \mathbb{E}\left(\left\|\sum_{m=1}^M \nabla f(x_{j-\tau_{j,m}})\right\|^2\right). \tag{29}$$

Summarizing the inequality (29) from $k = 1$ to $k = K$, we have

$$\mathbb{E}(f(x_{K+1})) - f(x_1)$$

$$\leq -\frac{M}{2}\sum_{k=1}^K \gamma_k \mathbb{E}\left(\|\nabla f(x_k)\|^2\right) + \sum_{k=1}^K \left(\frac{\gamma_k^2 L}{2} - \frac{\gamma_k}{2M}\right)\mathbb{E}\left(\left\|\sum_{m=1}^M \nabla f(x_{k-\tau_{k,m}})\right\|^2\right)$$

$$+ \sum_{k=1}^K \left(\frac{\gamma_k^2 ML}{2} + L^2 M^2 \gamma_k \sum_{j=k-T}^{k-1} \gamma_j^2\right)\sigma^2$$

$$+ L^2 MT \sum_{k=1}^K \left(\gamma_k \sum_{j=k-T}^{k-1} \gamma_j^2 \mathbb{E}\left(\left\|\sum_{m=1}^M \nabla f(x_{j-\tau_{j,m}})\right\|^2\right)\right)$$

$$= -\frac{M}{2}\sum_{k=1}^K \gamma_k \mathbb{E}\left(\|\nabla f(x_k)\|^2\right)$$

$$+ \sum_{k=1}^K \left(\gamma_k^2 \left(\frac{L}{2} + L^2 MT \sum_{\kappa=1}^T \gamma_{k+\kappa}\right) - \frac{\gamma_k}{2M}\right)\mathbb{E}\left(\left\|\sum_{m=1}^M \nabla f(x_{k-\tau_{k,m}})\right\|^2\right)$$

$$+ \sum_{k=1}^K \left(\frac{\gamma_k^2 ML}{2} + L^2 M^2 \gamma_k \sum_{j=k-T}^{k-1} \gamma_j^2\right)\sigma^2$$

$$\leq -\frac{M}{2}\sum_{k=1}^K \gamma_k \mathbb{E}\left(\|\nabla f(x_k)\|^2\right) + \sum_{k=1}^K \left(\frac{\gamma_k^2 ML}{2} + L^2 M^2 \gamma_k \sum_{j=k-T}^{k-1} \gamma_j^2\right)\sigma^2$$

where the last inequality is due to (7). Note that $x^*$ is the global optimization point. Thus we have

$$\frac{1}{\sum_{k=1}^K \gamma_k}\sum_{k=1}^K \gamma_k \mathbb{E}(\|\nabla f(x_k)\|^2) \leq \frac{2(f(x_1) - f(x^*)) + \sum_{k=1}^K \left(\gamma_k^2 ML + 2L^2 M^2 \gamma_k \sum_{j=k-T}^{k-1} \gamma_j^2\right)\sigma^2}{M\sum_{k=1}^K \gamma_k}.$$

It completes the proof. □

**Proofs to Corollary 2**

*Proof.* From (9) and (10), we have

$$\gamma \leq \frac{1}{2ML(T+2)}. \tag{30}$$

If follows that

$$\frac{1}{2}\gamma L + L^2 MT^2 \gamma^2 \leq \frac{1}{4M(T+2)} + \frac{T^2}{4M(T+2)^2} \leq \frac{1}{2M},$$

which implies that the condition (7) in Theorem 1 is satisfied globally. Then we can safely apply (8) in Theorem 1:

$$\frac{1}{K}\sum_{i=1}^K \mathbb{E}(\|\nabla f(x_i)\|^2) \leq \frac{2(f(x_1) - f(x^*)) + K\left(\gamma^2 ML + 2L^2 M^2 T\gamma^3\right)\sigma^2}{MK\gamma}$$

$$= \frac{2(f(x_1) - f(x^*))}{MK\gamma} + L\sigma^2\gamma + 2L^2 MT\sigma^2\gamma^2$$

$$\le \frac{2(f(x_1) - f(x^*))}{MK\gamma} + 2L\sigma^2\gamma$$

$$= 4\sqrt{\frac{(f(x_1) - f(x^*))L}{MK}}\sigma,$$

where the second last inequality is due to (30) and the last equality uses (9). It completes the proof. $\square$

**Proofs to Theorem 3**

*Proof.* From the Lipschitzisan gradient assumption (5), we have

$$f(x_{k+1}) \le f(x_k) - \gamma \left\langle \nabla_{i_k} f(x_k), \sum_{m=1}^{M} (G(\hat{x}_{k,m}; \xi_{k,m}))_{i_k} \right\rangle + \frac{L_{\max}\gamma^2}{2} \left( \sum_{m=1}^{M} (G(\hat{x}_{k,m}; \xi_{k,m}))_{i_k} \right)^2.$$
(31)

Taking the expectation of $i_k$ on both sides of (31), we have

$$\mathbb{E}_{i_k} f(x_{k+1}) \le f(x_k) - \frac{\gamma}{n} \left\langle \nabla f(x_k), \sum_{m=1}^{M} G(\hat{x}_{k,m}; \xi_{k,m}) \right\rangle + \frac{L_{\max}\gamma^2}{2n} \left\| \sum_{m=1}^{M} G(\hat{x}_{k,m}; \xi_{k,m}) \right\|^2.$$
(32)

Taking the expectation of $\xi_{k,*}$ on both sides of (32), we obtain

$$\mathbb{E}_{\xi_{k,*}, i_k} (f(x_{k+1}))$$

$$\le f(x_k) - \frac{\gamma}{n} \mathbb{E}_{\xi_{k,*}} \left[ \left\langle \nabla f(x_k), \sum_{m=1}^{M} G(\hat{x}_{k,m}; \xi_{k,m}) \right\rangle \right] + \frac{L_{\max}\gamma^2}{2n} \mathbb{E}_{\xi_{k,*}} \left[ \left\| \sum_{m=1}^{M} G(\hat{x}_{k,m}; \xi_{k,m}) \right\|^2 \right]$$

$$= f(x_k) - \frac{\gamma}{n} \left\langle \nabla f(x_k), \sum_{m=1}^{M} \nabla f(\hat{x}_{k,m}) \right\rangle + \frac{L_{\max}\gamma^2}{2n} \mathbb{E}_{\xi_{k,*}} \left[ \left\| \sum_{m=1}^{M} G(\hat{x}_{k,m}; \xi_{k,m}) \right\|^2 \right]$$

$$= f(x_k) - \frac{M\gamma}{2n} \left( \|\nabla f(x_k)\|^2 + \left\| \frac{1}{M} \sum_{m=1}^{M} \nabla f(\hat{x}_{k,m}) \right\|^2 - \underbrace{\left\| \nabla f(x_k) - \frac{1}{M} \sum_{m=1}^{M} \nabla f(\hat{x}_{k,m}) \right\|^2}_{T_1} \right)$$

$$+ \frac{L_{\max}\gamma^2}{2n} \mathbb{E}_{\xi_{k,*}} \underbrace{\left[ \left\| \sum_{m=1}^{M} G(\hat{x}_{k,m}; \xi_{k,m}) \right\|^2 \right]}_{T_2}$$
(33)

where the last equation is deduced by the fact $2\langle a, b \rangle = \|a\|^2 + \|b\|^2 - \|a - b\|^2$. We next consider $T_1$ and $T_2$ respectively.

For $T_2$, we have

$$\mathbb{E}_{\xi_{k,*}} (T_2) = \mathbb{E}_{\xi_{k,*}} \left[ \left\| \sum_{m=1}^{M} G(\hat{x}_{k,m}; \xi_{k,m}) \right\|^2 \right]$$

$$= \mathbb{E}_{\xi_{k,*}} \left[ \left\| \sum_{m=1}^{M} (G(\hat{x}_{k,m}; \xi_{k,m}) - \nabla f(\hat{x}_{k,m})) + \sum_{m=1}^{M} \nabla f(\hat{x}_{k,m}) \right\|^2 \right]$$

$$= \mathbb{E}_{\xi_{k,*}} \left[ \left\| \sum_{m=1}^{M} (G(\hat{x}_{k,m}; \xi_{k,m}) - \nabla f(\hat{x}_{k,m})) \right\|^2 \right.$$

$$+ \left\| \sum_{m=1}^{M} \nabla f(\hat{x}_{k,m}) \right\|^2 + 2 \left\langle \sum_{m=1}^{M} (G(\hat{x}_{k,m}; \xi_{k,m}) - \nabla f(\hat{x}_{k,m})), \sum_{m=1}^{M} \nabla f(\hat{x}_{k,m}) \right\rangle \right]$$

$$= \mathbb{E}_{\xi_{k,*}} \left[ \left\| \sum_{m=1}^{M} (G(\hat{x}_{k,m}; \xi_{k,m}) - \nabla f(\hat{x}_{k,m})) \right\|^2 + \left\| \sum_{m=1}^{M} \nabla f(\hat{x}_{k,m}) \right\|^2 \right]$$

$$=\mathbb{E}_{\xi_{k,*}}\left[\sum_{m=1}^{M}\|G(\hat{x}_{k,m};\xi_{k,m})-\nabla f(\hat{x}_{k,m})\|^2\right.$$

$$+2\sum_{1\le m<m'\le M}\langle G(\hat{x}_{k,m};\xi_{k,m})-\nabla f(\hat{x}_{k,m}),G(\hat{x}_{k,m'};\xi_{k,m'})-\nabla f(\hat{x}_{k,m'})\rangle$$

$$\left.+\left\|\sum_{m=1}^{M}\nabla f(\hat{x}_{k,m})\right\|^2\right]$$

$$\le M\sigma^2+\left\|\sum_{m=1}^{M}\nabla f(\hat{x}_{k,m})\right\|^2 \tag{34}$$

where the forth equality is due to

$$\mathbb{E}_{\xi_{k,*}}\left\langle\sum_{m=1}^{M}(G(\hat{x}_{k,m};\xi_{k,m})-\nabla f(\hat{x}_{k,m})),\sum_{m=1}^{M}\nabla f(\hat{x}_{k,m})\right\rangle$$

$$=\left\langle\sum_{m=1}^{M}\mathbb{E}_{\xi_{k,*}}(G(\hat{x}_{k,m};\xi_{k,m})-\nabla f(\hat{x}_{k,m})),\sum_{m=1}^{M}\nabla f(\hat{x}_{k,m})\right\rangle$$

$$=0$$

and the last inequality is due to the assumption (3) and

$$\mathbb{E}_{\xi_{k,*}}\sum_{1\le m<m'\le M}\langle G(\hat{x}_{k,m};\xi_{k,m})-\nabla f(\hat{x}_{k,m}),G(\hat{x}_{k,m'};\xi_{k,m'})-\nabla f(\hat{x}_{k,m'})\rangle$$

$$=\mathbb{E}_{\xi_{k,*}}\sum_{1\le m<m'\le M}\mathbb{E}_{k,m'}\langle G(\hat{x}_{k,m};\xi_{k,m})-\nabla f(\hat{x}_{k,m}),G(\hat{x}_{k,m'};\xi_{k,m'})-\nabla f(\hat{x}_{k,m'})\rangle$$

$$=\mathbb{E}_{\xi_{k,*}}\sum_{1\le m<m'\le M}\langle G(\hat{x}_{k,m};\xi_{k,m})-\nabla f(\hat{x}_{k,m}),\mathbb{E}_{k,m'}G(\hat{x}_{k,m'};\xi_{k,m'})-\nabla f(\hat{x}_{k,m'})\rangle$$

$$=0. \tag{35}$$

As for $T_1$, we have:

$$T_1=\left\|\nabla f(x_k)-\frac{1}{M}\sum_{m=1}^{M}\nabla f(\hat{x}_{k,m})\right\|^2$$

$$=\frac{1}{M^2}\left\|\sum_{m=1}^{M}(\nabla f(x_k)-\nabla f(\hat{x}_{k,m}))\right\|^2$$

$$\le\frac{1}{M}\sum_{m=1}^{M}\|(\nabla f(x_k)-\nabla f(\hat{x}_{k,m}))\|^2$$

$$\le\frac{L_T^2}{M}\sum_{m=1}^{M}\|x_k-\hat{x}_{k,m}\|^2$$

$$\le L_T^2\max_{k\in\{1,\cdots,M\}}\|x_k-\hat{x}_{k,m}\|^2$$

$$=L_T^2\|x_k-\hat{x}_{k,\mu}\|^2. \quad (\text{let }\mu:=\arg\max_{m\in\{1,\cdots,M\}}\|x_k-x_{k-\tau_{k,m}}\|^2)$$

It follows that

$$T_1\le L_T^2\|x_k-\hat{x}_{k,\mu}\|^2$$

$$=L_T^2\left\|\sum_{j\in J(k,\mu)}(x_{j+1}-x_j)\right\|^2$$

$$=L_T^2\gamma^2\left\|\sum_{j\in J(k,\mu)}\sum_{m=1}^{M}(G(\hat{x}_{j,m};\xi_{j,m}))_{i_j}e_{i_j}\right\|^2$$

$$=L_T^2\gamma^2\left\|\sum_{j\in J(k,\mu)}\sum_{m=1}^{M}[(G(\hat{x}_{j,m};\xi_{j,m}))_{i_j}-\nabla_{i_j}f(\hat{x}_{j,m})]e_{i_j}+\sum_{j\in J(k,\mu)}\sum_{m=1}^{M}\nabla_{i_j}f(\hat{x}_{j,m})e_{i_j}\right\|^2$$

$$\leq 2L_T^2\gamma^2 \left( \underbrace{\left\| \sum_{j\in J(k,\mu)} \sum_{m=1}^{M} \left[ (G(\hat{x}_{j,m};\xi_{j,m}))_{i_j} - \nabla_{i_j} f\left(\hat{x}_{j,m}\right) \right] e_{i_j} \right\|^2}_{T_3} + \underbrace{\left\| \sum_{j\in J(k,\mu)} \sum_{m=1}^{M} \nabla_{i_j} f\left(\hat{x}_{j,m}\right) e_{i_j} \right\|^2}_{T_4} \right)$$
(36)

where the last inequality uses the fact that $\|a+b\|^2 \leq 2\|a\|^2 + 2\|b\|^2$ for any real vector $a$ and $b$. Taking the expectation in terms of $i_j$ and $\xi_{j,*}$ with all $j$'s in $J(k,\mu)$ for $T_3$, we have

$$\mathbb{E}_{\xi_{j,*},i_j,j\in J(k,\mu)}(T_3)$$

$$=\mathbb{E}_{\xi_{j,*},i_j,j\in J(k,\mu)} \left( \left\| \sum_{j\in J(k,\mu)} \sum_{m=1}^{M} \left[ (G(\hat{x}_{j,m};\xi_{j,m}))_{i_j} - \nabla_{i_j} f(\hat{x}_{j,m}) \right] e_{i_j} \right\|^2 \right)$$

$$=\mathbb{E}_{\xi_{j,*},i_j,j\in J(k,\mu)} \left( \sum_{j\in J(k,\mu)} \left\| \sum_{m=1}^{M} \left[ (G(\hat{x}_{j,m};\xi_{j,m}))_{i_j} - \nabla_{i_j} f(\hat{x}_{j,m}) \right] e_{i_j} \right\|^2 \right)$$

$$+ \mathbb{E}_{\xi_{j,*},i_j,j\in J(k,\mu)} \left( \sum_{j''\neq j',j'',j'\in J(k,\mu)} \left\langle \sum_{m=1}^{M} \left[ (G(\hat{x}_{j'',m};\xi_{j'',m}))_{i_{j''}} - \nabla_{i_{j''}} f(\hat{x}_{j'',m}) \right] e_{i_{j''}}, \right.\right.$$

$$\left.\left. \sum_{m=1}^{M} \left[ (G(\hat{x}_{j',m};\xi_{j',m}))_{i_{j'}} - \nabla_{i_{j'}} f(\hat{x}_{j',m}) \right] e_{i_{j'}} \right\rangle \right)$$

$$=\mathbb{E}_{\xi_{j,*},i_j,j\in J(k,\mu)} \left( \sum_{j\in J(k,\mu)} \left\| \sum_{m=1}^{M} \left[ (G(\hat{x}_{j,m};\xi_{j,m}))_{i_j} - \nabla_{i_j} f(\hat{x}_{j,m}) \right] e_{i_j} \right\|^2 \right)$$

$$=\frac{1}{n}\mathbb{E}_{\xi_{j,*},j\in J(k,\mu)} \left( \sum_{j\in J(k,\mu)} \left\| \sum_{m=1}^{M} \left[ G(\hat{x}_{j,m};\xi_{j,m}) - \nabla f(\hat{x}_{j,m}) \right] \right\|^2 \right)$$

$$=\frac{1}{n}\mathbb{E}_{\xi_{j,*},j\in J(k,\mu)} \left( \sum_{j\in J(k,\mu)} \sum_{m=1}^{M} \| [G(\hat{x}_{j,m};\xi_{j,m}) - \nabla f(\hat{x}_{j,m})] \|^2 \right)$$

$$\leq \frac{TM\sigma^2}{n}.$$
(37)

where the second last equality is due to (35) and the third equality is due to

$$\mathbb{E}_{\xi_{j*},i_j,j\in J(k,\mu)} \left( \sum_{j''\neq j',j'',j'\in J(k,\mu)} \left\langle \sum_{m=1}^{M} \left[ (G(\hat{x}_{j'',m};\xi_{j'',m}))_{i_{j''}} - \nabla_{i_{j''}} f(\hat{x}_{j'',m}) \right] e_{i_{j''}}, \right.\right.$$

$$\left.\left. \sum_{m=1}^{M} \left[ (G(\hat{x}_{j',m};\xi_{j',m}))_{i_{j'}} - \nabla_{i_{j'}} f(\hat{x}_{j',m}) \right] e_{i_{j'}} \right\rangle \right)$$

$$=2\mathbb{E}_{\xi_{j*},i_j,j\in J(k,\mu)} \left( \sum_{j''>j',j'',j'\in J(k,\mu)} \left\langle \sum_{m=1}^{M} \left[ (G(\hat{x}_{j'',m};\xi_{j'',m}))_{i_{j''}} - \nabla_{i_{j''}} f(\hat{x}_{j'',m}) \right] e_{i_{j''}}, \right.\right.$$

$$\left.\left. \sum_{m=1}^{M} \left[ (G(\hat{x}_{j',m};\xi_{j',m}))_{i_{j'}} - \nabla_{i_{j'}} f(\hat{x}_{j',m}) \right] e_{i_{j'}} \right\rangle \right)$$

$$=2\mathbb{E}_{\xi_{j*},i_j,j\in J(k,\mu)} \left( \sum_{j''>j',j'',j'\in J(k,\mu)} \left\langle \sum_{m=1}^{M} \left[ (\mathbb{E}_{\xi_{j'',m}} G(\hat{x}_{j'',m};\xi_{j'',m}))_{i_{j''}} - \nabla_{i_{j''}} f(\hat{x}_{j'',m}) \right] e_{i_{j''}}, \right.\right.$$

$$\left.\left. \sum_{m=1}^{M} \left[ (G(\hat{x}_{j',m};\xi_{j',m}))_{i_{j'}} - \nabla_{i_{j'}} f(\hat{x}_{j',m}) \right] e_{i_{j'}} \right\rangle \right)$$

$$=0.$$

Taking the expectation in terms of $i_j$ with all $j$'s in $J(k,\mu)$ for $T_4$, we have

$$\mathbb{E}_{i_j,j\in J(k,\mu)}(T_4)$$

$$
=\mathbb{E}_{i_j, j \in J(k,\mu)} \left( \left\| \sum_{j \in J(k,\mu)} \sum_{m=1}^{M} \nabla_{i_j} f(\hat{x}_{j,m}) e_{i_j} \right\|^2 \right)
$$

$$
=\mathbb{E}_{i_j, j \in J(k,\mu)} \left[ \sum_{j \in J(k,\mu)} \left\| \sum_{m=1}^{M} \nabla_{i_j} f(\hat{x}_{j,m}) e_{i_j} \right\|^2 \right.
$$

$$
\left. + 2 \sum_{j'' > j', j'', j' \in J(k,\mu)} \left\langle \sum_{m=1}^{M} \nabla_{i_{j''}} f(\hat{x}_{j'',m}) e_{i_{j''}}, \sum_{m=1}^{M} \nabla_{i_{j'}} f(\hat{x}_{j',m}) e_{i_{j'}} \right\rangle \right]
$$

$$
=\mathbb{E}_{i_j, j \in J(k,\mu)} \left[ \frac{1}{n} \sum_{j \in J(k,\mu)} \left\| \sum_{m=1}^{M} \nabla f(\hat{x}_{j,m}) \right\|^2 \right.
$$

$$
\left. + 2 \sum_{j'' > j', j'', j' \in J(k,\mu)} \left\langle \sum_{m=1}^{M} \nabla_{i_{j''}} f(\hat{x}_{j'',m}) e_{i_{j''}}, \sum_{m=1}^{M} \nabla_{i_{j'}} f(\hat{x}_{j',m}) e_{i_{j'}} \right\rangle \right]
$$

$$
\leq \mathbb{E}_{i_j, j \in J(k,\mu)} \left[ \frac{1}{n} \sum_{j \in J(k,\mu)} \left\| \sum_{m=1}^{M} \nabla f(\hat{x}_{j,m}) \right\|^2 \right.
$$

$$
\left. + \frac{1}{n} \sum_{j'' > j', j'', j' \in J(k,\mu)} \left( \frac{1}{\alpha} \left\| \sum_{m=1}^{M} \nabla f(\hat{x}_{j'',m}) \right\|^2 + \frac{\alpha}{n} \left\| \sum_{m=1}^{M} \nabla f(\hat{x}_{j',m}) \right\|^2 \right) \right] \text{(Let } \alpha = \sqrt{n})
$$

$$
\leq \left( \frac{\sqrt{n} + T - 1}{n^{3/2}} \right) \sum_{j \in J(k,\mu)} \mathbb{E}_{i_j, j \in J(k,\mu)} \left( \left\| \sum_{m=1}^{M} \nabla f(\hat{x}_{j,m}) \right\|^2 \right), \tag{38}
$$

where the second last inequality is due to that for any $j'' > j'$:

$$
\mathbb{E}_{i_{j''}, i_{j'}} \left\langle \sum_{m=1}^{M} \nabla_{i_{j''}} f(\hat{x}_{j'',m}) e_{i_{j''}}, \sum_{m=1}^{M} \nabla_{i_{j'}} f(\hat{x}_{j',m}) e_{i_{j'}} \right\rangle
$$

$$
=\frac{1}{n} \mathbb{E}_{i_{j'}} \left\langle \sum_{m=1}^{M} \nabla f(\hat{x}_{j'',m}), \sum_{m=1}^{M} \nabla_{i_{j'}} f(\hat{x}_{j',m}) e_{i_{j'}} \right\rangle
$$

$$
\leq \frac{1}{n} \mathbb{E}_{i_{j'}} \left( \frac{1}{2\alpha} \left\| \sum_{m=1}^{M} \nabla f(\hat{x}_{j'',m}) \right\|^2 + \frac{\alpha}{2} \left\| \sum_{m=1}^{M} \nabla_{i_{j'}} f(\hat{x}_{j',m}) e_{i_{j'}} \right\|^2 \right)
$$

$$
=\frac{1}{n} \mathbb{E}_{i_{j'}} \left( \frac{1}{2\alpha} \left\| \sum_{m=1}^{M} \nabla f(\hat{x}_{j'',m}) \right\|^2 + \frac{\alpha}{2n} \left\| \sum_{m=1}^{M} \nabla f(\hat{x}_{j',m}) \right\|^2 \right).
$$

Take full expectation on both sides of (38), (37) and (36). Then substituting the upper bound of $\mathbb{E}(T_3)$ and $\mathbb{E}(T_4)$ into $\mathbb{E}(T_1)$, we have

$$
\mathbb{E}(T_1) \leq 2L_T^2 \gamma^2 \left( \frac{TM\sigma^2}{n} + \left( \frac{\sqrt{n} + T - 1}{n^{3/2}} \right) \sum_{j \in J(k,\mu)} \mathbb{E} \left( \left\| \sum_{m=1}^{M} \nabla f(\hat{x}_{j,m}) \right\|^2 \right) \right) \tag{39}
$$

Take full expectation on both sides of (34) and (33). Substituting $\mathbb{E}(T_1)$ and $\mathbb{E}(T_2)$ into (33), we have

$$
\mathbb{E}\left(f\left(x_{k+1}\right)\right) \leq \mathbb{E}\left(f\left(x_k\right)\right) - \frac{M\gamma}{2n} \left[ \mathbb{E}\left(\|\nabla f\left(x_k\right)\|^2\right) + \mathbb{E} \left( \left\| \frac{1}{M} \sum_{m=1}^{M} \nabla f\left(\hat{x}_{k,m}\right) \right\|^2 \right) \right.
$$

$$
- 2L_T^2 \gamma^2 \left( \frac{TM\sigma^2}{n} + \left( \frac{\sqrt{n} + T - 1}{n^{3/2}} \right) \sum_{j \in J(k,\mu)} \mathbb{E} \left( \left\| \sum_{m=1}^{M} \nabla f\left(\hat{x}_{j,m}\right) \right\|^2 \right) \right) \Bigg]
$$

$$
+ \frac{L_{\max}\gamma^2}{2n} \left( M\sigma^2 + \mathbb{E} \left( \left\| \sum_{m=1}^{M} \nabla f\left(\hat{x}_{k,m}\right) \right\|^2 \right) \right)
$$

$$
=\mathbb{E}\left(f\left(x_k\right)\right) - \frac{M\gamma}{2n} \mathbb{E}\left(\|\nabla f\left(x_k\right)\|^2\right) - \frac{\gamma}{2Mn} \mathbb{E} \left( \left\| \sum_{m=1}^{M} \nabla f\left(\hat{x}_{k,m}\right) \right\|^2 \right)
$$

$$+ \frac{M\gamma^3}{n} L_T^2 \left( \left( \frac{\sqrt{n} + T - 1}{n^{3/2}} \right) \sum_{j \in J(k,\mu)} \mathbb{E} \left( \left\| \sum_{m=1}^{M} \nabla f \left( \hat{x}_{j,m} \right) \right\|^2 \right) \right)$$

$$+ \frac{L_{\max}\gamma^2}{2n} \left( \mathbb{E} \left\| \sum_{m=1}^{M} \nabla f \left( \hat{x}_{k,m} \right) \right\|^2 \right) + \frac{L_T^2 TM^2 \gamma^3}{n^2} \sigma^2 + \frac{L_{\max} M \gamma^2}{2n} \sigma^2. \tag{40}$$

Summarizing (40) from $k = 1$ to $K$, we have

$$\mathbb{E}\left( f\left( x_{K+1} \right) \right) \leq f\left( x_1 \right) - \sum_{k=1}^{K} \frac{M\gamma}{2n} \mathbb{E}\left( \| \nabla f\left( x_k \right) \|^2 \right) - \frac{\gamma}{2Mn} \sum_{k=1}^{K} \mathbb{E} \left( \left\| \sum_{m=1}^{M} \nabla f \left( \hat{x}_{k,m} \right) \right\|^2 \right)$$

$$+ \frac{M\gamma^3}{n} L_T^2 \left( \left( \frac{\sqrt{n} + T - 1}{n^{3/2}} \right) \sum_{k=1}^{K} \sum_{j \in J(k,\mu)} \mathbb{E} \left( \left\| \sum_{m=1}^{M} \nabla f \left( \hat{x}_{j,m} \right) \right\|^2 \right) \right)$$

$$+ \frac{L_{\max}\gamma^2}{2n} \sum_{k=1}^{K} \left( \mathbb{E} \left( \left\| \sum_{m=1}^{M} \nabla f \left( \hat{x}_{k,m} \right) \right\|^2 \right) \right) + \frac{K L_T^2 TM^2 \gamma^3}{n^2} \sigma^2 + \frac{K L_{\max} M \gamma^2}{2n} \sigma^2$$

$$\leq f\left( x_1 \right) - \sum_{k=1}^{K} \frac{M\gamma}{2n} \mathbb{E}\left( \| \nabla f\left( x_k \right) \|^2 \right) - \frac{\gamma}{2Mn} \sum_{k=1}^{K} \mathbb{E} \left( \left\| \sum_{m=1}^{M} \nabla f \left( \hat{x}_{k,m} \right) \right\|^2 \right)$$

$$+ \frac{MT\gamma^3}{n} L_T^2 \left( \left( \frac{\sqrt{n} + T - 1}{n^{3/2}} \right) \sum_{k=1}^{K} \mathbb{E} \left( \left\| \sum_{m=1}^{M} \nabla f \left( \hat{x}_{k,m} \right) \right\|^2 \right) \right)$$

$$+ \frac{L_{\max}\gamma^2}{2n} \sum_{k=1}^{K} \left( \mathbb{E} \left( \left\| \sum_{m=1}^{M} \nabla f \left( \hat{x}_{k,m} \right) \right\|^2 \right) \right) + \frac{K L_T^2 TM^2 \gamma^3}{n^2} \sigma^2 + \frac{K L_{\max} M \gamma^2}{2n} \sigma^2$$

$$\leq f\left( x_1 \right) - \sum_{k=1}^{K} \frac{M\gamma}{2n} \mathbb{E}\left( \| \nabla f\left( x_k \right) \|^2 \right) + \frac{K L_T^2 TM^2 \gamma^3}{n^2} \sigma^2 + \frac{K L_{\max} M \gamma^2}{2n} \sigma^2$$

$$+ \left( \frac{MTL_T^2 \left( \sqrt{n} + T - 1 \right) \gamma^3}{n^{5/2}} - \frac{\gamma}{2Mn} + \frac{L_{\max}\gamma^2}{2n} \right) \sum_{k=1}^{K} \left( \mathbb{E} \left\| \sum_{m=1}^{M} \nabla f \left( \hat{x}_{k,m} \right) \right\|^2 \right)$$

$$\leq f\left( x_1 \right) - \sum_{k=1}^{K} \frac{M\gamma}{2n} \mathbb{E}\left( \| \nabla f\left( x_k \right) \|^2 \right) + \frac{K L_T^2 TM^2 \gamma^3}{n^2} \sigma^2 + \frac{K L_{\max} M \gamma^2}{2n} \sigma^2$$

where the last inequality comes from (15). Together with $\mathbb{E}(f(x_{k+1})) \geq f(x^*)$, we have

$$\frac{1}{K} \sum_{t=1}^{K} \mathbb{E} \| \nabla f(x_t) \|^2 \leq \frac{2n}{KM\gamma} \left( f(x_1) - f(x^*) \right) + \frac{2L_T^2 TM\gamma^2}{n} \sigma^2 + L_{\max} \gamma \sigma^2. \tag{41}$$

It completes the proof. $\qquad \square$

**Proofs to Corollary 4**

*Proof.* From the definition of the steplength (17) and the lower bound of $K$ (18), we have

$$\gamma = \sqrt{\frac{2n(f(x_1) - f(x^*))}{L_T M \sigma^2}} \frac{1}{\sqrt{K}}$$

$$\leq \sqrt{\frac{2n(f(x_1) - f(x^*))}{L_T M \sigma^2}} \sqrt{\frac{\sqrt{n}\sigma^2}{16(f(x_1) - f(x^*)) L_T M \left( n^{3/2} + 4T^2 \right)}}$$

$$= \frac{1}{2} \sqrt{\frac{1}{2n^{3/2} + 8T^2}} \frac{n^{3/4}}{L_T M} \tag{42}$$

which gives an upper bound for $\gamma$. We can further relax this upper bound by

$$\gamma \leq \frac{1}{2} \sqrt{\frac{1}{2n^{3/2} + 8T^2}} \frac{n^{3/4}}{L_T M} \leq \frac{1}{2} \sqrt{\frac{1}{8T^2}} \frac{n^{3/4}}{L_T M} < \frac{2n}{L_T M T}. \tag{43}$$

Next we will show that the steplength satisfies the condition in (15):

$$
\begin{aligned}
\text{LHS} &= \frac{2M^2TL_T^2\left(\sqrt{n}+T-1\right)\gamma^2}{n^{3/2}} + 2ML_{\max}\gamma \\
&= \frac{2M^2TL_T^2\left(\sqrt{n}+T-1\right)}{n^{3/2}}\frac{1}{4}\frac{1}{2n^{3/2}+8T^2}\frac{n^{3/2}}{L_T^2M^2} + \frac{L_{\max}M}{2}\sqrt{\frac{1}{2n^{3/2}+8T^2}}\frac{n^{3/4}}{L_TM} \\
&\leq \frac{1}{2}\frac{T\left(\sqrt{n}+T\right)}{n^{3/2}+4T^2} + \frac{1}{2} \\
&\leq \frac{1}{2}\frac{T\sqrt{n}+T^2}{n^{3/2}+4T^2} + \frac{1}{2} \\
&\leq \frac{1}{2}\frac{3T^2}{n^{3/2}+4T^2} + \frac{1}{2}\frac{2n}{n^{3/2}+4T^2} + \frac{1}{2} \\
&\leq \frac{1}{2}\cdot\frac{3}{5} + \frac{1}{2}\cdot\frac{2}{5} + \frac{1}{2} \\
&\leq 1 = \text{RHS},
\end{aligned}
$$

where the first inequality uses the fact $L_{\max} \leq L_T$ and the first inequality uses the upper bound of $\gamma$ in (42). The third last inequality comes from $\sqrt{n}T \leq 2n + 2T^2$. It means that the condition (15) in Theorem 3 is satisfied globally. Then we can safely apply (16) in Theorem 3:

$$
\begin{aligned}
\frac{1}{K}\sum_{t=1}^{K}\mathbb{E}\left\|\nabla f\left(x_t\right)\right\|^2 &\leq \frac{2n}{KM\gamma}\left(f\left(x_1\right)-f\left(x^*\right)\right) + \frac{2L_T^2TM\gamma^2}{n}\sigma^2 + L_{\max}\gamma\sigma^2 \\
&\leq \frac{2n}{KM\gamma}\left(f\left(x_1\right)-f\left(x^*\right)\right) + 5L_T\gamma\sigma^2 \\
&= \frac{6\sqrt{2\left(f\left(x_1\right)-f\left(x^*\right)\right)L_T}n\sigma}{\sqrt{KM}}
\end{aligned}
$$

where the second inequality is due to the upper bound of $\gamma$ in (43) and the last equality is acquired by substituting $\gamma$ by its definition in (17). It completes the proof. $\qquad\square$

## Footnotes

[5] https://github.com/BVLC/caffe

[6] http://yann.lecun.com/exdb/mnist/

[7] https://www.mpich.org/