[Reviews · NeurIPS 2015]

Submitted by Assigned_Reviewer_1

The authors provide the improved analysis of the convergence rate of parallel stochastic gradient algorithms. They show the gradient will decrease with a rate of 1/sqrt{K} under mild condition on objective function and bounded delay, and provide the analysis for both distributed and shared-memory asynchronous SG.

This is a good paper and I enjoy reading it. It is well known that parallel SG is useful in practice,

and surprisingly the analysis was pretty limited and the guarantee was only for convex functions.

The authors provide an improved convergence analysis, and even extend to the nonconvex functions, which is very exciting for me. I just have the following minor suggestions/comments:

- In line 296 (Section 4), is the "Update" step using atomic writes? Otherwise there might be multiple writes to the same x, and I wonder whether this is modeled in the analysis. The choices of atomic/non-atomic writes have been discussed in the following paper for parallel coordinate descent:

PASSCoDe: Parallel ASynchronous Stochastic dual Co-ordinate Descent. In ICML 2015.

- eq (5): a parenthesis is misplaced.

Summary: A good paper. The authors provide an improved convergence analysis of parallel asynchronous SG, and their analysis can be applied to nonconvex functions.

Submitted by Assigned_Reviewer_2

The authors consider distributed (synchronous), and parallel (asynchronous) minibatch stochastic gradient methods for smooth problems, without the assumption of convexity. The main contribution is a generalization of the bounds of Agarwal and Duchi, and Niu et al [Hogwild!], for the case of mini-batch SGM, and the restriction to smooth functions (not necessarily convex).

In this work, convergence is defined with respect to the expected norm of the sampled gradients. They show that in both setups (distributed sync., or parallel async.) stochastic gradient can achieve linear speedups as long as the number of cores is smaller than the square root of the number of iterations (this is a O(T^1/4) improvement over the works of Agarwal and Duchi, for the sync case).

Unfortunately, the paper is not well-written (many typos, and stylistic issues, some listed below), and although the subject matter is very interesting, the technical contribution is not particularly novel and is of limited importance. On one hand the authors analyze previously known algorithms using mini-batches, but the results that they obtain (using smoothness) are not insightful for real applications. It is not clear for example whether there is any novel intuition provided in this paper on why these algorithms work in practice. Also, it is not clear why saddle point (ergodic) convergence is relevant to the analysis of learning NNs (the authors use that to motivate their analysis).

Some Typos/Stylistic comments:

eq 9: shouldn't that be max?

A comment that is inaccurate "our analysis considers the smooth nonconvex optimization which covers the smooth strongly convex optimization considered in their analysis; " In the Niu et al. paper, the convergence rate is 1/T (as the authors consider strongly convex functions). It is not clear how the presented results of 1/sqrt(T) rates contain the Niu et al. result as a case.

- "one is on the computer network and the other is on the shared memory system. " -> "one is over a computer network and the other is on a shared memory system."

- the following phrases are too informal:

"people still know very little about its properties in nonconvex optimization"

"People even have no idea if its convergence is certified for nonconvex optimization"

- "Our analysis provides theoretical support and guarantee" -> Our analysis provides theoretical guarantees

- "A parallel algorithm to stochastic gradient" not sure what the authors mean here (a dual algorithm?)

- "Please refer to the online learning literatures" -> Please refer to the online learning literature

"Ghadimi and Lan [2013] proves" -> Ghadimi and Lan [2013] prove

Summary: The authors consider distributed (synchronous), and parallel (asynchronous) minibatch stochastic gradient methods for smooth problems, not assuming convexity. Although the subject matter is very interesting, the novelty of the technical contribution is limited.

Submitted by Assigned_Reviewer_3

The paper gives assumptions under which two known asynchronous descent algorithms, AsySG-Con and Asy-Incon, demonstrate ergodic convergence rates that hold for non-convex objective functions in the stochastic optimization setting. Specifically, they show that when the number of iterations is quadratic in the delay time, then O(1/\sqrt{MK}) convergence is possible. Moreover, they also derive more favorable upper bounds for AsySG-Con in the stochastic convex optimization setting than those previously known.

Quality: The paper is well-organized, from surveying recent work in work in asynchronous stochastic optimization to clearly stating their assumptions and results. They also provide some experiments re-validating the known success of the above algorithms.

Clarity: It is not clear why ergodic convergence rate is a useful metric for analyzing non-convex objective functions and optimization problems. It doesn't seem to reveal anything about local optima and/or saddle points, which are some of the main theoretical points of contention for deep learning (one of the main examples that the authors cite for motivation).

Originality: Although the algorithms are well-known, to the best of the reviewer's knowledge, the convergence results are new.

Significance: While the improvement in the convex scenario is nice, as mentioned in the clarity section, the authors do not do a good job convincing the reader that their result is useful or insightful in the non-convex setting. Since they are studying existing algorithms, this is a very important point.

Edit: In the authors' rebuttal, they provided a couple of references for ergodic convergence rates and promised to elaborate on their usefulness for locating saddle points in the final draft of their paper. However,

it is still not clear that convergence to arbitrary saddle points is a good enough reason for why gradient descent algorithms (and their asynchronous counterparts) perform well for NN-type objectives.
Summary: The authors prove new results for two well-known algorithms, but do not provide a good job justifying the importance of their guarantees in the main non-convex setting of their paper.

Submitted by Assigned_Reviewer_4

The paper establishes a theoretical analysis for the asynchronous parallel stochastic gradient algorithms for nonconvex optimization. The paper is well written and well motivated. For the ASYSG-CON algorithm, the authors provided a better upper bound for the maximal number of workers ensuring the linear speed up than previous work (Theorem 1). For the ASYSG-INCON algorithm, the authors provided a convergence analysis without strongly convex assumption and consistent read requirement, which is eligible for more scenarios. The experiments validate the speedup property of their analysis by using deep neural network based models.

I have a few questions about this paper:

1. In both Theorem 1 and Theorem 2, the step size \gamma is dependent on the global optimal solution x^*. However, we don't know the value of x^* initially. How to choose \gamma in practice to evaluate the theoretical result? The authors let \gamma=1.1e-7 for ASYSG-INCON in experiments. How to obtain this value? And what about the step size for the experiments of ASYSG-CON?

2. I notice that the authors evaluated ASYSG-CON on well known dataset LENET and CIFAR, but the experiments of ASYSG-INCON only include the synthetic data. Is there any result for ASYSG-INCON on the real-world dataset?
Summary: The paper establishes a theoretical analysis for the asynchronous parallel stochastic gradient algorithms for nonconvex optimization. The paper is well written and well motivated.

Submitted by Assigned_Reviewer_5

The authors extended the stochastic gradient and coordinate descent to the asynchronous and nonconvex differentiable setting, assuming that the asynchronism is bounded. The authors showed that the average of the expected gradient converges to 0 at the usual sublinear rate O(1/sqrt{K}), where K is the number of iterations. The proofs appear to be solid and the authors did a fair job in comparing with relevant works in the literature. Very limited experiments were given in the appendix for a quick sanity check. Overall, the results in this work may be a good addition to the field.

Quality: The proofs mostly appear to be correct, and the model, much built on previous work, is useful and relevant in practice. However, I am not quite convinced by the measure of convergence: the average of the expected gradients converge to 0 seems to be too weak. First, what can we conclude from the expectation here? If we run the algorithm for say 10 repetitions, we cannot really average the iterates to get a single iterate that has a strong guarantee. This is the cost of not having convexity. Second, the obtained results do not cover the known facts for convex functions. It would be much more convincing had one been able to include the special convex case.

Some technical issues:

1). In assumption 3, i_k is allowed to be dependent on xi_k. This seems to contradict Eq. (33) in line 937, i.e., conditioned on xi_k, i_k may not be uniformly distributed. This may invalidate the claim in Eq. (16) (line 406). 2). Line 814, here one should first condition on all xi_s, s < j'' (for fixed m), and then take expectation wrt xi_{j''} to zero out the term involving xi_{j''}. This is because x_{j''-tau_{j'',m}} may depend on xi_{j'} so we cannot first take expectation wrt xi_{j'}. (Also, here \nabla_{i_{j'}} should be \nabla, multiple times.) 3). Some calculations from line 1228 to 1236 seems to be messed up, but it does not seem to affect the claim. 4). The conditions in theorem 1 and 2 both involve f(x^*). This is undesirable. It would be great if one can come up with practical, usable bounds.

Clarity: The paper is mostly easy to follow.

Originality: The authors mostly follow the existing model in the literature but extend them to the nonconvex setting. The proofs do not involve any new techniques. So overall the work is incremental, but I think it is still of some interest.

Significance: I feel that the authors oversell their work a bit, for the obtained results are far from "explaining deep learning or nonconvex stochastic optimization." The obtained sublinear rates also need more careful interpretation.

Reference: The following work also seems to be relevant and perhaps should be cited: Ho et al., More Effective Distributed ML via a Stale Synchronous Parallel Parameter Server. NIPS 2013.
Summary: In this work the authors extended the asynchronous stochastic gradient and coordinate descent to the nonconvex setting, and proved some O(1/sqrt{K}) sublinear "convergence" rate, where K is the number of iteration. The proof seems to be solid although I have some doubt on the appropriateness of the "convergence" criteria (see below). Considering the recent interest in nonconvex optimization and parallel computing, this work can be a good addition to the field.

Submitted by Assigned_Reviewer_6

the paper is relatively easy to follow, literature review is sufficient.

I very like the paper as it breaches the gap between the experience from using SGD algorithms in practice and the theory.

In (5) there is a typo, e_i on left hand side should be close to \alpha_i and not after ")".

Summary: In this paper two versions of asynchronous mini-batch SDG method are analyzed. In contrast of many related papers (e.g. HogWild!), in this work they do not assume that the objective function is convex. They show that the linear speed-up can be achieved up to \sqrt{K} delay.

Author Feedback
Author rebuttal: We thank all reviews for their constructive comments and suggestions.

To All:

A common question asked by R1, R2 and R7 is how the ergodic convergence (rate) is related to the convergence (rate) in the common sense and how it indicates the convergence to the saddle point for explaining the convergence of AsySG in solving NNs or general nonconvex optimization. First we want to point out that ergodic convergence is a commonly used metric in the analysis in optimization, especially for nonconvex optimization (for example, gradient descent [pp. 27, Eq. (1.2.14) Nesterov's textbook, 2004] and stochastic gradient [Ghadimi and Lan, 2013]). Based on existing studies, the (sublinear) ergodic convergence rate immediately implies the following two consequences. Let {q_k = E(||F'(x_k)||^2)} denote the expected gradient sequence. 1) One can construct a subsequence which converges in the same rate by using the trick in [S. Ghadimi and Lan 2013, see Eqs. (2.2) and (2.3)]. The subsequence is constructed by randomly selecting an iterate (say, x_{r(k)}) from previous k iterates, and then E(q_{r(k)}) = E(||F'(x_{r(k)})||^2) <= O(1/k^0.5). Therefore, this result suggests that the constructed subsequence {x_{r(k)}} converges to the saddle point with high probability (using Markov inequality). 2) The ergodic convergence rate also implies that the limit inferior convergence liminf_k q_k <= O(1/k^0.5). We will include these important implications in our revision to illustrate the connection with convergence (rate) to the saddle point for NNs or general nonconvex optimization.

R1:
Intuition of AsySG: While the SG (a single thread) uses the "stochastic" gradient to surrogate the accurate gradient, the asynchronous implementation of SG (AsySG) brings additional deviation from the accurate gradient due to using "stale" (or delayed) information. If the deviation due to "staleness" is relatively minor to the deviation due to "stochastic" in SG, the iteration complexity (or convergence rate) of AsySG is comparable to SG, which implies the nearly linear speedup. All conditions in Theorems 1 and 2 are essentially used to restrict the "staleness". That is the key reason why AsySG works. We will highlight this insight in our revision.

This paper does not claim any technical contribution. We only use existing optimization analysis tools as well as stealing many smart ideas from existing work such as hogwild! [Niu et al. 2011], AsySCD [Liu and Wright, 2014], Agarwal and Duchi [2011], S. Ghadimi and G. Lan [2013], Li et al. [2014], and many others.

This work is relevant to the analysis of learning NNs in the following sense: the saddle point convergence explains the convergence of widely used AsySG for solving NNs since in general NNs is the nonconvex optimization.

Thanks for pointing out those typos and grammars.

R2:
Thanks for your comment. Please read To All.

R3:
Thanks for your support on our paper.

R4:
We assume that the update on a single component is atomic (it is guaranteed by most modern architectures as pointed out in Hogwild!) but not for updating the whole vector. We also noticed the PASSCoDe paper after we submitted our paper. We will appropriately cite the missing paper.

R5:
Thanks for your support and constructive comments. We will consider including more details and intuitions of experiments by appropriately shortening Section 4.

R6:
For fair measurement on speedup, the steplength in AsySG-INCON is set as the best steplength for a single thread (that is equivalent to the SG). The steplength length used in AsySG-CON is set to be the recommended steplength for each particular data set by Caffe, which can be considered as the optimal choice for serial SG. The main purpose of experiments is to validate the speedup property, but it does not hurt to include additional experiments for AsySG-Incon on real data in the full version of this paper.

R7:
Thanks for your deep review on this paper. It is hard to recover the convergence rate for general convex objectives from the analysis for nonconvex case due to the difficulty of bounding ||x-x*|| or f(x) - f* by ||F'(x)||. But for a little bit more restricted convex objectives, for example, functions with error bounds or optimal strongly convex functions (both are supersets of strongly convex functions), our analysis is able to recover the convergence rate for the convex case.

As for the technical issues you pointed out, all of them can be easily fixed: 1) To make it more clear, we plan to substitute the sampling scheme in Step 3 of Algorithm 2 with the more general version in Step 3 of Algorithm 3; 2) Yes, you are right. Actually, it is a typo. We should take exception as you suggested order; 3) Yes, it can be fixed without changing the claim; 4) Indeed, the term f(x*) in our theorems can be replaced by any lower bound of it (which is easy to obtain in practice).

We will appropriately include the missing literature in our revision.